# Changes in groundwater drought associated with anthropogenic warming

John P. Bloomfield[1], Benjamin P. Marchant[2], Andrew A. McKenzie[1]

[1] British Geological Survey, Maclean Building Crowmarsh Gifford, Oxfordshire, OX10 8BB, UK

[2] British Geological Survey, Environmental Science Centre, Keyworth, Nottinghamshire, NG12 5GD, UK

*Correspondence to*: John P. Bloomfield (jpb@bgs.ac.uk)

**Abstract.** Here we present the first empirical evidence for changes in groundwater drought associated with anthropogenic warming in the absence of long-term changes in precipitation. Analysing standardised indices of monthly groundwater levels, precipitation and temperature, using two unique groundwater level data sets from the Chalk aquifer, UK, for the period 1891 to 2015, we show that precipitation deficits are the main control on groundwater drought formation and propagation. However, long-term changes in groundwater drought are shown to be associated with anthropogenic warming over the study period. These include increases in the frequency and intensity of individual groundwater drought months, and increases in the frequency, magnitude and intensity of episodes of groundwater drought, as well as an increasing tendency for both longer episodes of groundwater drought and for an increase in droughts of less than one year in duration. We also identify a transition from coincidence of episodes of groundwater drought with precipitation droughts at the end of the 19th century, to an increasing coincidence with both precipitation droughts and with hot periods in the early 21st century. In the absence of long-term changes in precipitation deficits, we infer that the changing nature of groundwater droughts is due to changes in evapotranspiration (ET) associated with anthropogenic warming. We note that although the water tables are relatively deep at the two study sites, a thick capillary fringe of at least 30m in the Chalk means that ET should not be limited by precipitation at either site. ET may be supported by groundwater through major episodes of groundwater drought and, hence, long-term changes in ET associated with anthropogenic warming may drive long-term changes in groundwater drought phenomena in the Chalk aquifer. Given the extent of shallow groundwater globally, anthropogenic warming may widely effect changes to groundwater drought characteristics in temperate environments.

## 1 Introduction

Globally groundwater provides of the order of one third of all freshwater supplies (Doll et al, 2012; 2014), 2.5 billion people are estimated to depend solely on groundwater for basic daily water needs (UN, 2015), and it sustains the health of many important groundwater-dependent terrestrial ecosystems (Gleeson et al, 2012). This high level of dependence on groundwater means that communities and ecosystems across the globe are vulnerable to both natural variations in groundwater resources (Wada et al., 2010) and to the impacts of anthropogenic climate change on groundwater (Green et al., 2011; Taylor et al., 2013). Groundwater droughts, taken here to mean periods of below normal groundwater levels (Tallaksen & Van Lanen, 2004; Van Loon, 2015; Van Loon et al., 2016a; 2016b), are a major threat to global water security (Van Lanen et al., 2013) and are potentially susceptible to being modified by climate change. Since drought formation and propagation are

driven by precipitation deficits and evapotranspirative losses associated with elevated temperatures, there is an
expectation that anthropogenic climate change, and in particular anthropogenic warming is already modifying
the occurrence and nature of droughts (Dai, 2011; Sheffield et al., 2012; Trenberth et al., 2014; Greve et al.,
2014). However, to date there have been no systematic investigations of how warming due to anthropogenic
climate change may affect groundwater drought (Green et al., 2011; Taylor et al., 2013; Jackson et al., 2015),
and it has even been noted in the IPCC Fifth Annual Assessment of Impacts, Adaptation, and Vulnerability that:
'there is no evidence that … groundwater drought frequency has changed over the last few decades' (Jiménez
Cisneros et al., 2014, p.232). This significant gap in our understanding of groundwater drought is not surprising
given the limited availability of long groundwater level time series suitable for analysis (Jimenez Cisneros et al.,
2014) and the low signal to noise ratios characteristic of many hydrological systems (Wilby 2006; Watts et al.,
2015). In addition, the challenges of formal attribution of groundwater droughts due to anthropogenic warming
(Trenberth et al., 2015), and the potentially confounding influences of land use change and groundwater
abstraction on groundwater drought (Stoll et al., 2011; Jimenez Cisneros et al., 2014; Van Loon et al., 2016a;
2016b) complicate any analysis of such droughts. Here we address some of these challenges and present the first
empirical evidence for the effects of anthropogenic warming on the changing nature groundwater droughts.

Groundwater systems have been shown to effect global land-energy budgets and regional climate (Senevirante
et al., 2006; Trenberth et al., 2009; Maxwell and Condon, 2016) and can control the generation of large-scale
hydrological droughts, particularly in temperate climates (Van Lanen et al., 2013). Although the role of
evapotranspiration (ET) in regional- to global-scale drying is still a matter of active debate (Dai, 2011; Sheffield
et al., 2012; Greve et al., 2014; Milly and Dunne, 2016), there is an expectation of a general increase in ET, and
hence of general drying associated with anthropogenic warming (Trenberth et al., 2014). Even if anthropogenic
warming may not necessarily cause droughts, Trenberth et al. have noted that it is expected that when droughts
do occur that they are likely to set in more quickly and to be more intense in a warming world (Trenberth et al.,
2014). In order to investigate the evidence for such changes in groundwater droughts it is desirable to identify
sites from unconfined aquifers with long, continuous records of groundwater levels where there have been no
systematic long-term changes in precipitation or land cover. In addition, the sites ideally should be free from the
systematic, long-term influence of groundwater abstraction. In this context, we investigate the empirical
evidence for changes in the character of groundwater droughts in the period 1891 to 2015 associated with
anthropogenic warming at two such sites, representative of groundwater systems in temperate climates, from the
Cretaceous Chalk, the major aquifer of the UK. These sites are at Chilgrove House (CH), believed to be the
world's longest continuously monitored groundwater level observation borehole, and at Dalton Holme (DH)
(Figure 1).

We have adopted an approach similar to that of Diffebaugh et al. (2015) in order to investigate how
anthropogenic warming may have effected groundwater droughts at CH and DH. Diffebaugh et al., (2015)
demonstrated how anthropogenic warming has increased hydrological drought risk in California over the last
approximately 100 years by comparing the changing frequency of drought, as measured by the Palmer Modified
Drought Index (PMDI), with standardised annual average precipitation and temperature anomalies. Here,
instead of using the PMDI, we use the Standardised Groundwater level Index, SGI (Bloomfield and Marchant,
2013) to characterise the monthly status of groundwater, and compare changes in SGI with changes in
standardised monthly air temperature and precipitation. We have chosen not to use the Standardised
Precipitation Evapotranspiration Index, SPEI, (Vincente-Serrano SM et al., 2010; Trenberth et al., 2014) in our
analysis as we explicitly wish to analyse the correlations between SGI and standardised temperature and
between SGI and standardised precipitation independently (Stagge et al., 2017).

We have not attempted to formally attribute any groundwater droughts to climate change. Rather, we follow the
approach of Trenberth et al. (2015) and investigate how climate change may modify a particular phenomenon of
interest. In our case, given the known centennial-scale anthropogenic warming over the UK described in section
2.2 (Sexton et al., 2004; Karoly and Stott, 2006; Jenkins et al., 2008), using an empirical analysis we address the
following question. How has the occurrence, duration, magnitude and intensity of groundwater drought, as
expressed by changes in monthly SGI and in episodes of groundwater drought, changed over the period 1891 to
2015? Once relationships between naturally varying precipitation anomalies, groundwater droughts and
anthropogenic warming are quantified and characterised, subsequent studies may consider attribution of
groundwater droughts. Such studies may address questions related to assessing how much of the anomaly in any
given groundwater drought can be explained by anthropogenic warming, but attribution-based questions are out
of scope of the current empirical study. Note also that investigation of the relationship, if any, between episodes
of extreme heat (heatwaves) and groundwater droughts is not in the scope of the present study. Although the
analysis is restricted to data from two sites in the UK, the findings have potentially significant implications for
changes in groundwater drought driven by anthropogenic warming given the global extent of shallow
groundwater systems (Fan et al., 2013) and this is discussed in section 5.

**2 Site descriptions & drought context**
The Chilgrove House (CH) and Dalton Holme (DH) sites meet the requirements of the study in that continuous,
long records of groundwater level are available from small rural catchments negligibly impacted by land-use
change and abstraction over the period of study (section 2.1). Importantly, both sites are subject to long-term
warming associated with anthropogenic climate change (section 2.2). In addition, there are no major long-term
changes in mean precipitation at the two sites over the analysis period (demonstrated qualitatively in section 4
and quantitatively through the results of a simple statistical test, section 4.1). The sites, although unusual due to
their length, continuous nature and frequency (monthly or better) of the groundwater level observations, are
representative of groundwater systems and hydrological settings that are common throughout large, populous
areas of the globe including Europe, Asia, N. and S. America and parts of Australia and southern Africa, in that
they represent shallow, unconfined aquifers in temperate regions.
**2.1 Site descriptions**
The CH and DH groundwater observation boreholes are located in the Chalk, the principal aquifer in the UK
(Downing et al., 1993; Lloyd, 1993). CH is in south-east England and DH in the east of England (Figure 1). The
CH and DH hydrographs are the two longest, continuous records in the UK's National Groundwater Level
Archive (NGLA) (British Geological Survey, 2017). Hydrographs such as these in the NGLA are taken to be
representative of the major UK aquifers, in this case of the Chalk aquifer, and were selected to being in areas
least affected by abstraction (Jackson et al., 2015). There are no major groundwater abstractions in the
immediate vicinity of the observation boreholes. Both observation boreholes are located in small rural
catchments with no major population centres or industrial activities, and there has been no long-term change in
land use in the catchments over the study period. The inference of a lack of any systematic impacts from
abstraction on groundwater levels at CH and DH is supported by the observed good correlation between
precipitation and groundwater levels at the two sites (Bloomfield and Marchant, 2013).

CH is a 62.0 m deep observation borehole in the Seaford Chalk Formation, a white chalk (limestone) of
Coniacian to Santonian age. Groundwater levels over the observation record have an absolute range of about
43.7 m, with a maximum groundwater level of about 77.2 meters above sea level (masl) and a minimum level of
about 33.5 masl, equivalent to <1 m to about 43 m below ground level at the site (Supplementary Information,
Figure S1). The hydrograph generally has an annual sinusoidal response, typical of unconfined Chalk, with an
annual groundwater fluctuation of around 25 m, although double and higher multiple recharge peaks and
episodes are also relatively frequent due to natural variability in precipitation (recharge can occur in summer as
well as winter with appropriate antecedent conditions and precipitation). Variability in winter recharge means
that in some winters, such as 1854-5, 1897-8, 1933-4, 1975-6, 1991-2 and 1995-6, minimal recharge occurs.
This characteristically results in a nearly continuous decline in groundwater levels throughout the recharge
season, and in all cases groundwater droughts occurred during the following summers. There are no clear
geological or catchment constraints on either the lowest or highest groundwater levels at CH. However, the
River Lavant, approximately 2 km to the south-east of the CH borehole, is a Chalk bourne stream that drains the
catchment. The flowing length and discharge of the Lavant reflect the regional groundwater level. Land cover in
the Lavant catchment is approximately 35% woodland, 65% arable and grassland with a small number of
villages. Comparison of Ordnance Survey land cover mapping from 1898 and 2015 shows that there has been no
substantial change in land cover in the vicinity of CH during this period (Ordnance Survey, 1897; 2015).

DH is a 28.5-m-deep observation borehole in the Burnham Chalk Formation, a thinly-bedded white chalk of
Turonian to Santonian age. Groundwater levels have a range of 14.8 m, with a maximum groundwater level of
about 23.8 masl and a minimum of about 9.6 masl, equivalent to a range from about 10 m to about 25 m below
ground level (Supplementary Information, Figure S1). The groundwater level hydrograph has a broadly
sinusoidal appearance. Groundwater levels at DH respond more slowly to rainfall than at CH, despite the thinner
unsaturated zone. This is probably due to local effects of thin glacial till deposits near DH. Maximum
groundwater levels at DH may be controlled by the elevation of springs that feed a small surface drain about
0.75 km to the south. There are no clear geological or catchment constraints on the lowest groundwater levels.
DH is located in a flat lying area with no major streams or rivers. Land cover in the immediate vicinity of DH is
predominantly arable and grassland with a number of small areas of woodland and villages. Comparison of
Ordnance Survey land cover mapping from 1892 and 2015 shows that there has been no substantial change in
land cover during this period (Ordnance Survey, 1911; 2015).

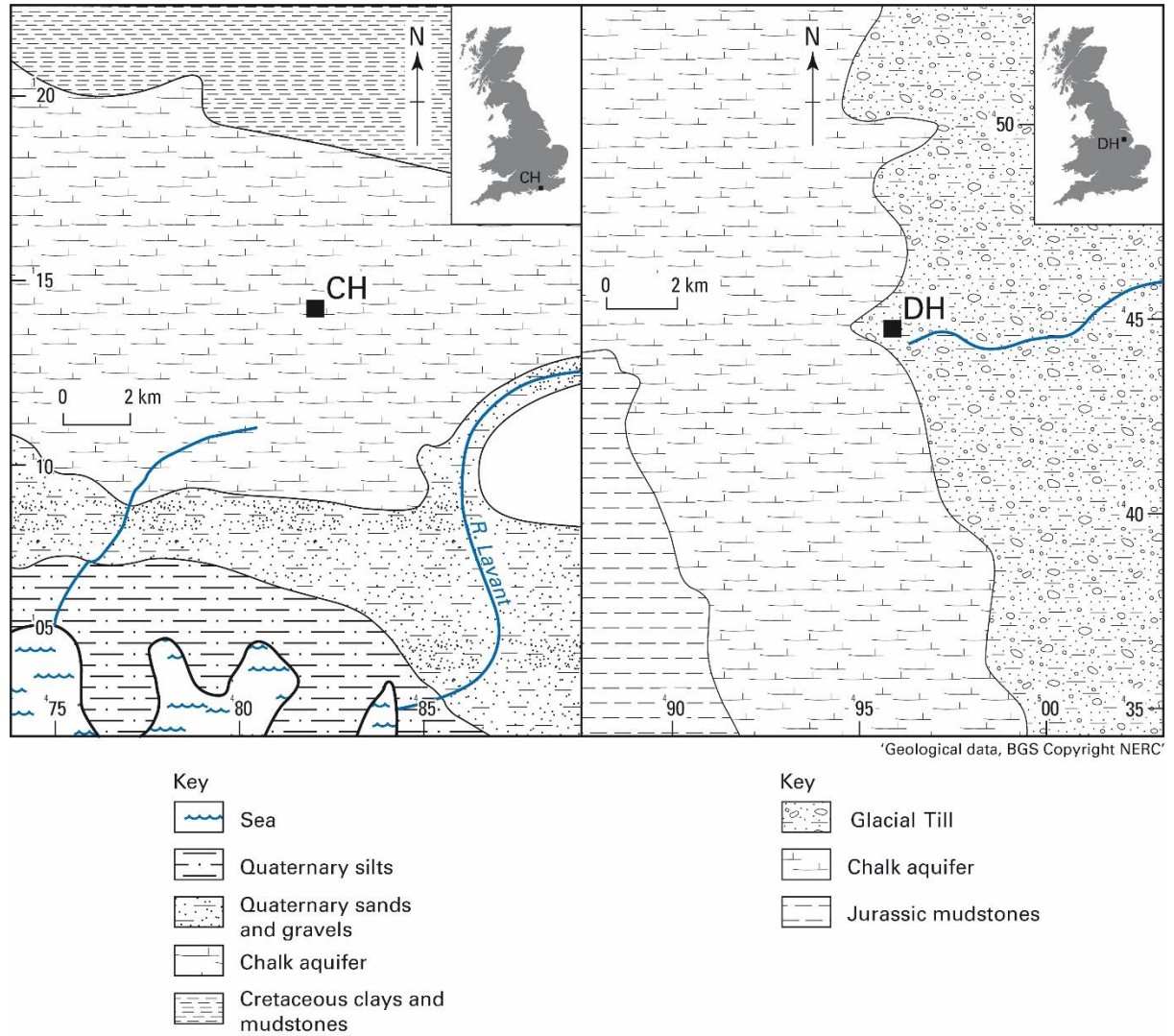


**Figure 1. Location of the CH and DH observation boreholes, local geological setting, coastline and surface water**
**courses.**
**2.2 Climate & drought context**
Average monthly air temperature at CH over the observation record from 1891 to 2015 is 9.4°C and at DH it is
9.1°C, with maximum and minimum average monthly temperatures of 19.8°C and 18.8°C and -3°C and -1.6°C
at CH and DH respectively (Supplementary Information, Figure S2). In the Köppen–Geiger classification (Peel
et al., 2007), the climate at CH and DH can be characterised as temperate and falls in the ocean or maritime
climate category, being representative of large parts of north-west Europe. Mean monthly precipitation at CH is
83 mm, slightly higher than at DH where the mean monthly precipitation is 58 mm.

Air temperature across both catchments closely follows the Central England Temperature (CET) monthly series
(Parker et al., 1992) (Supplementary Information, Figure S2). Analysis of the CET record shows that near-
surface air temperature in England has been rising at a rate of 0.077°C/decade since 1900 (Parker & Horton,
2005) and by about 0.42°C/decade between 1975 and 2005 (Karoly and Stott, 2006).  The observed recent
warming in mean annual CET at least since 1950 has been formally attributed to anthropogenic forcing (Sexton
et al., 2004; Karoly and Stott, 2006; Jenkins et al., 2008; King et al., 2015). In contrast, annual mean
precipitation over England shows no systematic trends since records began in 1766, and there has been no
attribution of changes in annual mean precipitation to anthropogenic factors (Jenkins et al., 2008; Watts et al.,
2015). In addition, we note that there is no evidence at either of the study sites for a systematic change in long-
term monthly precipitation across the observational records (see sections 3.2 and 4.1). Precipitation in the UK is,
however, seasonal and highly variable, with a tendency towards drier summers in the south-east and wetter
winters in the north-west (Jenkins et al., 2008; Watts et al., 2015), and the precipitation time series at CH and
DH show seasonal and inter-annual variations, including episodes of meteorological drought consistent with the
broad drought history of southern and eastern England (Supplementary Information, Figure S3).

A number of studies have described major episodes of hydrological drought, including groundwater drought, in
the UK since the 19th Century (Marsh et al., 2007; Lloyd-Hughes et al., 2010; Bloomfield & Marchant, 2013;
Bloomfield et al., 2015; Folland et al., 2015; Marchant and Bloomfield, 2018) and the societal impacts of those
droughts (Taylor et al., 2009; Lange et al., 2017). Marsh et al. (2007) identified seven episodes of major
hydrological droughts in England and Wales between 1890 and 2007 using ranked rainfall deficiency time series
and analysis of long river flow and groundwater level time series (Marsh et al., 2007, Table 2) as follows: 1890-
1910 (known as the 'Long Drought'); 1921-1922; 1933-34; 1959; 1976; 1990-92; and 1995-1997. Marsh et al.
(2007) noted that of these major droughts all but one, the drought of 1959, had sustained and/or severe impacts
on groundwater levels. All the major droughts typically had large geographical footprints extending over much
of England and Wales as well as over parts of north western Europe (Lloyd-Hughes and Saunders, 2002; Lloyd-
Hughes et al., 2010; Fleig et al., 2011; Hannaford et al., 2011). However, regional variations in drought
intensities were present within and between the major drought events as function of spatial differences in
driving meteorology and catchment and aquifer properties (Marsh et al., 2007; Bloomfield & Marchant, 2013;
Bloomfield et al., 2015; Marchant and Bloomfield, 2018).

Three periods have been used for analysis in this study, namely: 1891-1932, 1933-1973 and 1974-2015 (see
section 3.2). We note that this means that each analysis period contain episodes of previously documented major
historic drought. For example, the first analysis period includes the 'Long Drought' of 1890-1910 (Marsh et al.,
2007), the middle period includes the drought of 1933-1934, the most intense groundwater drought on record at
CH, and the last period includes the 1975-1976 drought a major groundwater drought at both CH and DH
(Bloomfield & Marchant, 2013).

**3 Data & methods**

**3.1 Data**

Groundwater level measurements are available back to 1836 for the observation borehole at CH, while
groundwater levels are available for DH back to 1889. We have chosen to analyse the 125-year-long series of
observed monthly groundwater levels from 1891 to 2015 common to both sites. The groundwater level data
have been taken from the National Groundwater Level Archive, NGLA (National Groundwater Level Archive,
2017). Groundwater level observations are typically at least at monthly intervals. Groundwater levels have been
linearly interpolated to the end of the month prior to standardisation, as the SGI requires groundwater level data
on a regular time step.

There are no continuous rain gauge or temperature records for either of the sites that cover the entire study
period. However, gridded temperature and precipitation data (5 km by 5 km gridded UKCP09 data) are
available for the UK for the period 1910 to 2011 (Met. Office, 2017). The work described here is part of a
Natural Environment Research Council (NERC) funded Historic Droughts project (see Acknowledgements) that
has extend these records back to 1891 using newly recovered, digitised and gridded meteorological records, and
extended the gridded data to 2015 using unpublished updates. We have used this extended gridded dataset in the
present study.  Although the extent of the groundwater catchments are unknown at both sites they are likely to
be restricted to a few square kms given the nature of the local hydrogeology (Figure 1). Consequently,
temperature and precipitation data have been extracted for the 5 km by 5 km grid cell in which the CH and DH
observation boreholes are located, in a manner analogues to Jackson et al., (2016), rather that averaging the
gridded climatological data over a larger area.
**3.2 Methods**
As mentioned in the introduction, the empirical analysis described in this paper broadly follows the approach of
Diffebaugh et al., (2015), i.e. analysis of changes in the frequency and co-occurrence of three standardised
indices with time, where one index is a measure of the hydrological drought status of the system and the other
two indices separately characterise precipitation and air temperature anomalies. Diffebaugh et al., (2015) chose
to analyse their 100-year-long records in two halves. However, in this study we have divided the observation
record into thirds to give three periods for use in the analysis, namely 1891-1932, 1933-1973 and 1974-2015.
This provides some granularity in the description of changes in the standardised indices with time and means
that the first period is associated with the least anthropogenic warming while that the last period, 1974-2015,
coincides with the period of greatest documented anthropogenic warming over the study area (Karoly and Stott,

231  2006).


An alternative approach to dividing the records for analysis could have been to try and identify one or more
significant change points in the temperature record using time series analysis techniques and then to use those
periods to characterise any differences in the relationships between hydrological droughts and features of the
driving climatology between those periods. Change point analysis (Chen & Gupta, 2000) can lack statistical
power because of the temporal correlation amongst the data and the need for a correction to account for the
multiple hypotheses that are in effect being considered. So for example, the Bonferroni correction (Bonferroni,
1936) would require that to demonstrate significance at the $p = 0.05$ level that each hypothesise be tested at the
$p = 0.05/1499 = 3 \times 10^{-5}$ level based on the length of the time series in the present study. When a change point
analyses was conducted on the monthly standardised groundwater level, precipitation and air temperature time
series from each site and the model residuals were assumed to be independent, significant steps were identified
in each series. However, when temporal correlation in the time series was accounted for with a first order auto-
regressive model, only steps in the air temperature series from both sites and the groundwater level series from
DH persisted. The most significant step in the CH air temperature series was in November 1988 ($p = 2 \times 10^{-7}$).
For the DH temperature series the most significant step was also in November 1988 ($p = 2 \times 10^{-8}$), and for the
DH groundwater level series it was in April 1984 ($p$ = 0.01). However, after a Bonferroni correction only the
steps in the monthly standardised air temperature time series remained significant.

Notwithstanding the results of the change point tests, the change point approach to defining analysis periods has
not been adopted in the present study for a couple of reasons. Since the aim of the study is to characterise
changes in relationships between groundwater droughts and climatology in the context of previously
documented long-term warming we want to make no prior assumptions regarding specific periods with
potentially different temperature regimes. In addition, as we know from previous studies that anthropogenic
warming will have effected both series since at least the 1950s (Sexton et al., 2004; Karoly and Stott, 2006;
Jenkins et al., 2008; King et al., 2015) the meaning of any change points identified post 1950 in the context of
anthropogenic warming would be unclear and is an approach towards attribution that we are explicitly trying to
avoid in the current study. However, we note that the change point analysis described above is consistent with
the findings presented in the Results (section 4), in that the latter part of the observational record at both sites is
significantly warmer than the earlier part of the record.

A wide range of methods have been used to characterise and investigate hydrological droughts including
groundwater droughts. They broadly fall into two classes: standardised indices and threshold level approaches
(see Van Loon, 2015 for a detailed recent overview). Threshold level approaches use a pre-defined threshold,
which may vary seasonally. When flows or, in the case of groundwater, when levels fall below a given threshold
a site is considered to be in drought. Drought characteristics, such as duration, magnitude and frequency can
then be estimated. This approach has the benefit of being able to characterise aspects of droughts in absolute
terms and hence is particularly useful for water resource management planning or to understand processes at a
particular observation borehole. However, it does not lend itself so easily to studies where there is a need to
compare multiple sites and multiple indicators of change. For example, in the present study it would be
necessary to define and justify six seasonally varying thresholds (two for each site identifying precipitation and
groundwater drought thresholds and one for each site identifying hot period thresholds). In contrast, droughts
characterised using standardisation approaches enable the comparison of hydrological anomalies between
different sites and/or between different components of the terrestrial water cycle using common standardised
anomalies from a normal situation (Van Loon, 2015). Given the need in this study to compare relative changes
in groundwater droughts at two sites across long observational records and to explore the relationships between
groundwater droughts and precipitation deficits and air temperature, we have chosen to use standardised drought
indices. This approach has the additional benefit of only needing to estimate a single common, consistent,
internationally recognised (WMO, 2012) descriptor of drought based on a standardised drought index.

The Standardised Groundwater level Index, SGI (Bloomfield and Marchant, 2013), has been estimated across
the full observational records from 1891 to 2015. It has been used to characterise monthly status of groundwater,
and to compare changes in SGI with changes in standardised monthly air temperature and precipitation over the
same period. The SGI builds on the Standardised Precipitation Index (SPI) of McKee et al. (1993) to account for
differences in the form and characteristics of groundwater level time series. The SPI was proposed by McKee et
al. (1993) as an objective precipitation-based measure of the severity and duration of meteorological droughts. It
assumes that drought status is described by a normally distributed index. However, Bloomfield and Marchant
(2013) demonstrated that parametric transformations of groundwater levels typically produced poor
approximations to normal distributions, and concluded that it is doubtful if the resulting standardised series
could be objectively compared. Instead Bloomfield and Marchant (2013) recommended a non–parametric
approach to the standardisation of groundwater level hydrographs similar to other non-parametric approaches,
for example Osti et al. (2008) who used a plotting position method to estimate a standardised precipitation, and
Vidal et al. (2010) who used a non-parametric kernel density fitting routine to estimate a normalised soil
moisture index.

SGI has been estimated using the monthly groundwater level time series. The SGI relies on a non–parametric
approach to the standardisation of groundwater level hydrographs (Bloomfield and Marchant, 2013).  It is
estimated using a normal-scores transform (Everitt, 2002) of groundwater level data for each calendar month.
This nonparametric normalisation assigns a value to observations, based on their rank within a data set, in this
case groundwater levels for a given month from a given hydrograph. The normal scores transform is undertaken
by applying the inverse normal cumulative distribution function to $n$ equally spaced $p_i$ values ranging from $1/(2$
$n)$ to $1-1/(2\ n)$. The values that result are the SGI values for the given month. These are then re-ordered such
that the largest SGI value is assigned to the $i$ for which $p_i$ is largest, the second largest SGI value is assigned to
the $i$ for which $p_i$ is second largest, and so on. The SGI distribution which results from this transform will
always pass the Kolmogorv–Smirnov test for normality. The normalisation is undertaken for each of the 12
calendar months separately and the resulting normalised monthly indices then merged to form a continuous SGI
time series. Note that the resulting standardised drought index is not linear and that drought conditions (SGI <-
1) will be expected about 16% of the time while extreme drought conditions (SGI <-2) would be expected only
about 2% of the time (McKee et al., 1993).

A Standardised Temperature Index (STI) and a Standardised Precipitation Index (SPI) have been calculated by
applying the SGI method to the average monthly temperature (STI) and a monthly accumulated rainfall (SPI)
time series across the full observational records from 1891 to 2015. Due to the lagged response of groundwater
levels to driving meteorology (Bloomfield and Marchant, 2013; Van Loon, 2015), correlations between SGI and
$STI_q$ and between SGI and $SPI_q$ will vary with $q$, where $q$ is the averaging period (for preceding months
temperature) or accumulation period (for preceding months precipitation). In order to assess changes in
groundwater droughts in the context of the driving climatology in a consistent manner, we estimate Pearson
cross-correlation coefficients between SGI and STI and between SGI and SPI for periods $q = 1$ to 12, and then
search for the period $q$ with the highest absolute summed cross-correlation (Supplementary Information, Figure
S4), i.e. the period that is associated with the highest correlation between groundwater levels and the antecedent
driving meteorology (both precipitation and temperature).

The maximum absolute summed cross-correlation for accumulation and averaging periods was found to be six
months at both sites, where individual cross-correlations between SGI and $SPI_6$ are 0.76 and 0.75. Note no
systematic variation is observed in the correlations between SGI and SPI and between SGI and STI (Figure S4)
across the observation record: correlations between SGI and SPI are similar in the first and last third of the
observational record. As would be expected, the cross-correlation between SGI and STI is weaker than that of
SGI and SPI with correlations between SGI and $STI_6$ of -0.15 and -0.35 for CH and DH respectively (Figure
S4). The six month maximum absolute summed cross-correlation period is consistent with previous analyses of
SPI accumulation at the two sites. It is the same as the SPI accumulation period identified by Bloomfield and
Marchant (2013) for CH and slightly less than that for DH (Bloomfield and Marchant, 2013, Table 2) (note that
cross-correlation co-efficients at DH are particularly insensitive to $q$ beyond six months, Figure S4). This is
despite the accumulation periods in Bloomfield and Marchant (2013) being based on standardised precipitation
alone and being estimated for a shorter observation record than the present study.

Given the above, we have used a six month accumulation for precipitation and a six month average for
temperature in the analysis. For simplicity, throughout the following description of the results and discussions,
all subsequent references to SPI and STI relate to $SPI_6$ and $STI_6$ unless otherwise stated. Although the
correlations between SGI, SPI and STI are based on simple phenomenological correlations between the time
series they reflect recharge and discharge processes at the sites and are consistent with accepted
conceptualisations of drought generation in the Chalk. For example, at both sites the standardised groundwater
level at the end of the winter recharge season, i.e. SGI in March, is correlated with accumulated precipitation for
the six months prior to March, i.e. the winter half year from October to March. Previously, Folland et al., (2015)
have documented a variety of climate and other drivers of multi-annual hydrological droughts across the English
Lowlands, the region within which CH and DH are located, based on precipitation deficits established during
winter half-years.

There is a plethora of definitions of meteorological drought (Lloyd-Hughes, 2014; Van Loon, 2015). Here we
follow the WMO convention for SPI (McKee et al., 1993; World Meteorological Organisation, 2012) where
precipitation drought is defined as any period of continuously negative SPI that reaches an intensity of -1 or less.
By analogy, we define any month with a negative SPI or SGI of -1 or less as a precipitation or groundwater
drought month and any month with a positive STI that reaches an intensity of 1 or more as a hot month. Periods
of continuously negative SGI or SPI that reaches a monthly intensity of -1 or less is defined as an episode of
groundwater ($SGI_e$) or precipitation ($SPI_e$) drought, and a period of continuously positive STI that reaches a
monthly intensity of 1 or more, denoted by $STI_e$, is defined as a hot period.

In addition, we are interested in the degree of co-incidence between episodes of groundwater drought,
precipitation drought, and hot periods. Groundwater droughts have been assessed to be co-incident with
precipitation droughts or with hot periods if both the following conditions are met: i.) any part of a groundwater
or precipitation drought episode or hot period overlaps, and ii.) within the periods, incidents of monthly SGI ≤ -
1 either overlap with or postdate incidents of either monthly SPI is ≤ -1, or monthly STI is ≥ 1.
**4 Results**
**4.1 Changes in standardised monthly temperature, groundwater level and precipitation since 1891**
SGI time series show that there has been a large increase in the frequency of months of groundwater drought
since 1891 at both sites (Figures 2 and 3 and Supplementary Information Table S1). The frequency of months of
groundwater drought has more than doubled between the first third (1891-1932) and the last third (1974-2015)
of the record, i.e. from about 10% of months at both sites in the first third of the record to 21% and 25% at CH
and DH respectively in the last third of the record. The increase in frequency of groundwater drought months is
associated with a very large increase in the frequency of hot months, from 5% to 34% of the months at CH and
from 2% to 34% of the months at DH. In contrast, there has been no systematic change in the frequency of
precipitation drought months. The probability of these changes in standardised indices between the first and last
thirds of the observational record being significant has been estimated. Given that the standardised monthly
indices are normally distributed, a null model can be estimated where each standardised index is assumed to be a
realisation of temporally auto-correlated Gaussian random function (with auto-correlation function estimated
from the observed data). A 'probability of difference' for a standardised index between analysis periods can be
estimated as follows. If we define D as equal to the number of droughts in the last analysis period minus the
number of droughts in the first analysis period (for example) then the probability of difference is the probability,
under the null model, that D is greater than the observed value. Estimated in this way, the probability of the
difference in the number of hot months in the period 1891-1932 and 1974-2015 being as extreme as the
observed is 0.03 for CH and 0.005 for DH. For groundwater drought months the probabilities are 0.056 for CH
and 0.055 for DH, but for precipitation drought months they are 0.70 for CH and 0.36 for DH. From this it is
inferred that the increased incidence of hot months and of groundwater drought months between the start and
end of the record is very unlikely to occur by chance at both sites, but that there is no significant difference in
the probability of precipitation drought.

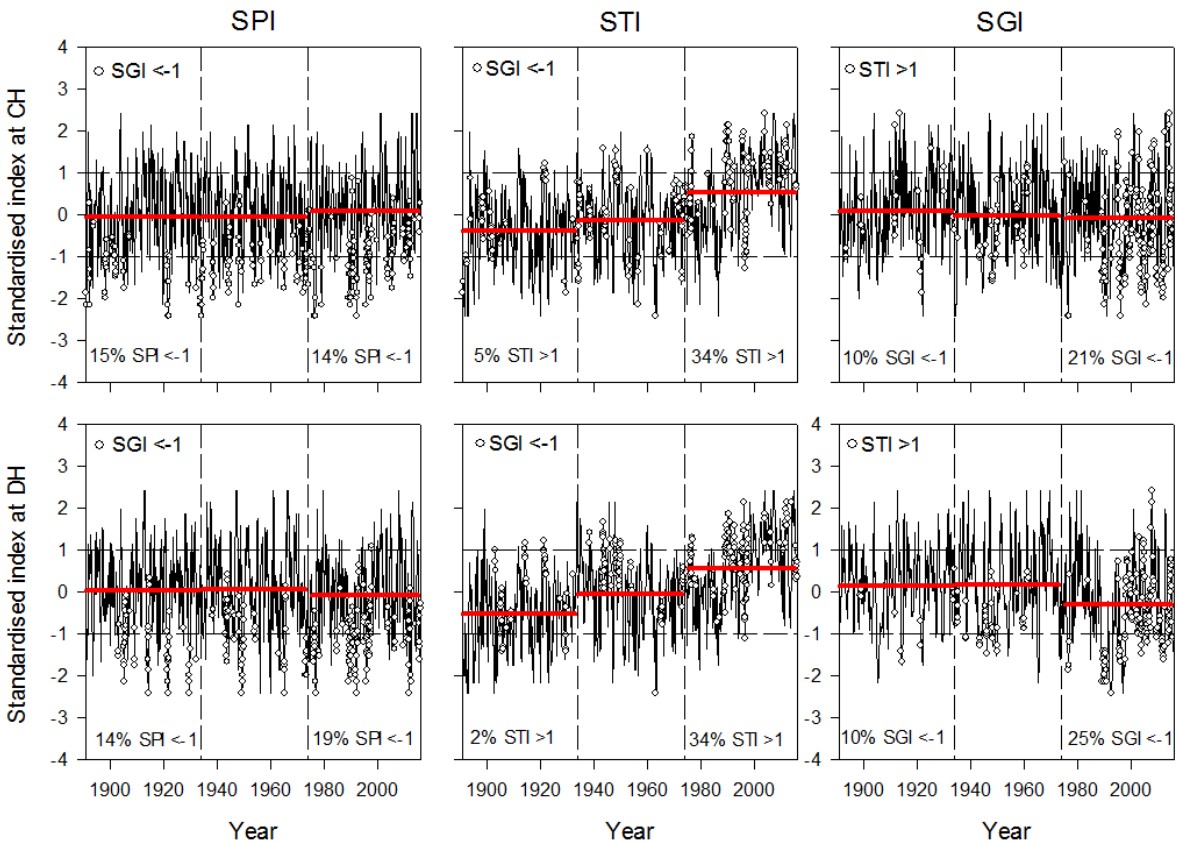

**Figure 2. Changes in standardised indices of precipitation, temperature and groundwater level since 1891. Time**
**series of SPI, STI, and SGI for CH (upper panels) and DH (lower panels) for the period 1891-2015, with mean values**
**for first, middle and last thirds of the record highlighted in red. Open circles in plots of SGI denote months where**
**STI is ≥ 1, and in plots of SPI and STI denote months where SGI is ≤ -1. Percentages are for months in the first and**
**last third of the records where SGI and SPI are ≤ -1 and STI is ≥ 1.**

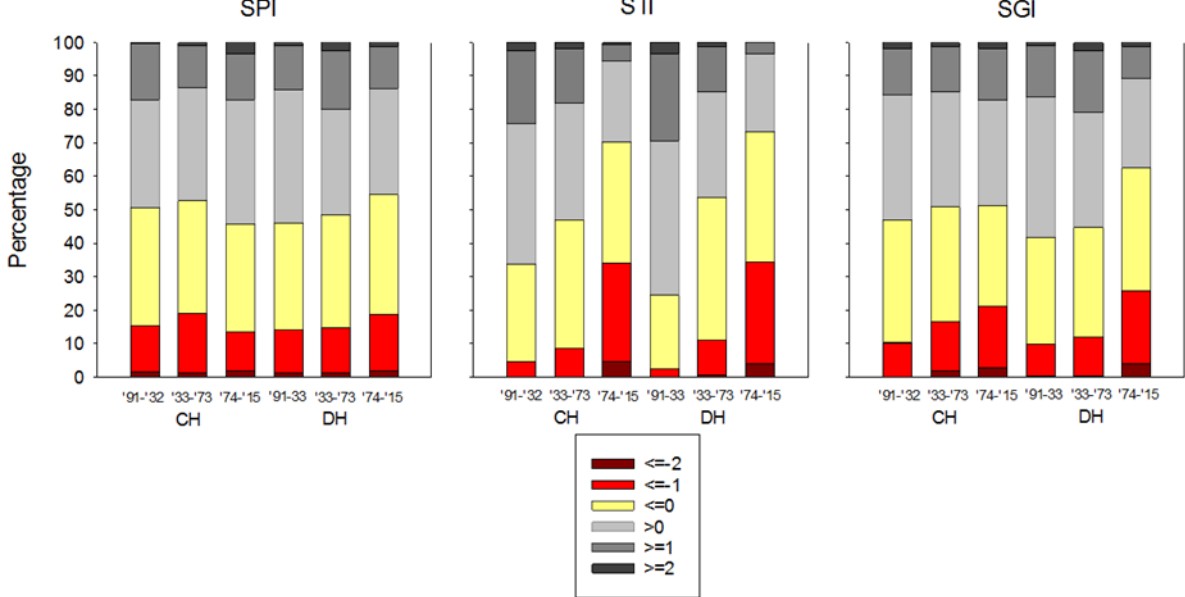


**Figure 3. Percentage of monthly SPI, STI and SGI as a function of six ranges of standardised values from ≤ -2 to ≥ 2**
**for each third of the records from CH and DH.**
**4.2 Changes in association between monthly groundwater drought, temperature and precipitation**
Figure 4 shows the occurrence of groundwater drought months as a function of SPI and STI at CH and DH for
the periods 1890-1932, 1933-1973 and 1974-2015. It shows how groundwater drought months reflect both
natural variability in the driving drought climatology, specifically in precipitation deficits, but also underlying
changes associated with anthropogenic warming.

Dry months (SPI <=0) appear to be a broad prerequisite for groundwater drought months throughout the record
at both sites (Figure 4). Natural variation in precipitation deficit is clearly the primary control on groundwater
drought months across the whole period at both sites and major groundwater droughts driven by significant
precipitation deficits are evident in the data.  For example, six of the eight most extreme groundwater drought
months in the middle period at CH (where SGI <-2) are all associated with a single episode of major
precipitation deficit and drought that lasted from Autumn 1933 to Autumn 1934 (see Supplementary
Information Figure S5). The drought of 1933-34 was related to consecutive dry summer and winter half years
resulting in effectively no groundwater recharge over a 12 month period (Marsh et al. 2007; Alexander & Jones,
2000). Because that drought occurred at the start of the middle third of the observational record it is associated
with a relatively cool period with respect to the full standardised series.

However, number of trends can be seen in addition to the natural variability in precipitation deficits. As a
consequence of the increase in frequency of groundwater drought months and of hot months across the
observational record, there has been a considerable increase in the number of groundwater drought months that
coincide with hot months, particularly in the last third of the record (Supplementary Information Table S2). The
percentage of groundwater drought months that coincided with hot months increased from about 8% and 10% in
the period 1891-1932, to about 48% and 43% in the period 1974-2015 for CH and DH respectively. At the same
time, there has been a slight reduction in the coincidence of months of groundwater and precipitation drought
for these two periods, from 67% to 48% and from 66% to 51% for CH and DH. For the last third of the record,
since 1974, this means that groundwater drought months are now almost as likely to coincide with hot months
as they are to coincide with months of precipitation drought. So the increase in the percentage of groundwater
drought months that coincide with both hot months and precipitation drought months between the first and last
periods of the observational record, from about 8% to 34% and from about 10% to 35% for CH and DH, is
almost entirely due to the effect of warming. To illustrate and emphasise the combined effects of these changes
in the relationship between SPI, STI and SGI across the three periods, Figure 4 shows the centroid for all
groundwater drought months (SGI<-1) for each third of the record. At CH and DH there is a strong overall
warming trend with mean STI for all groundwater drought months increasing from -0.16 to -0.04 to 0.85 at CH
and from 0.03 to 0.44 to 0.98 at DH between 1891-1932, 1933-1973 and 1974-2015. This is consistent with the
warming trend across the whole record (Figure 2 and 3).

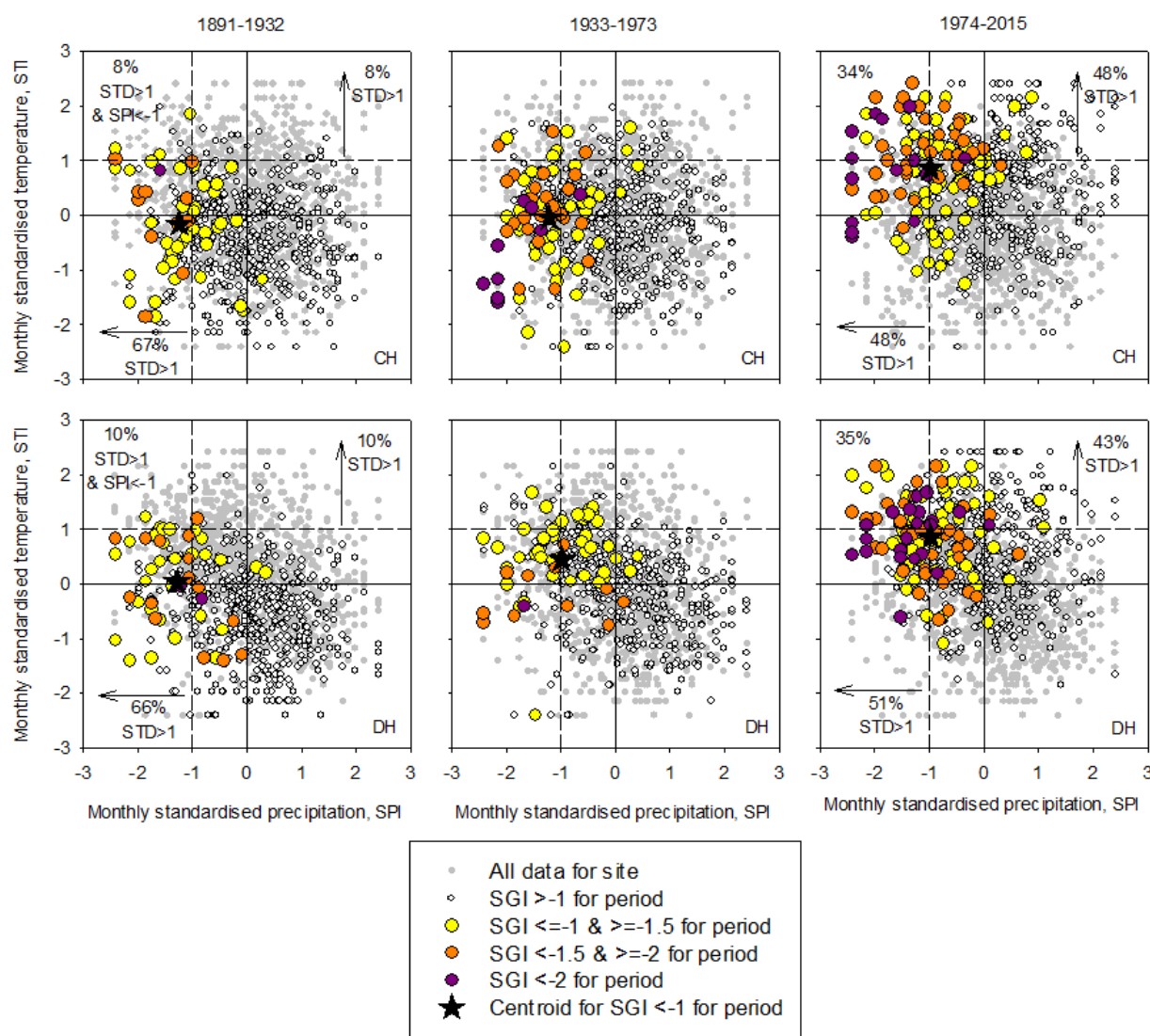


**Figure 4. Occurrence of groundwater drought months as a function of SPI and STI at CH (upper panels) and at DH (lower panels), for the periods 1890-1932, 1933-1973 and 1974-2015.  For reference, all monthly values from 1891-2015 are shown as grey symbols and all months were SGI >-1 for a given period are shown as open white circles. Black stars denote the centroid of the groundwater drought months for each of the three periods.**

## 4.3 Changes in episodes of groundwater drought

There have been 45 episodes of groundwater drought at CH and 33 at DH between 1891 and 2015. Of these, 16 episodes at CH and 9 episodes at DH had an average SGI intensity of ≤ -1 (Supplementary Information, Table S3). Across the observational record, groundwater droughts are more frequent but typically shorter and less intense at CH compared with DH (Figures 5 and 6). This is consistent with previous observations that the CH hydrograph has a shorter autocorrelation than that of DH, inferred to be due to differences in aquifer and catchment characteristics between the sites (Bloomfield and Marchant, 2013).

Episodes of groundwater drought are present throughout the observation record and are primarily driven by episodes of precipitation deficit. Episodes of groundwater drought ($SGI_e$) are almost entirely associated with dry episodes where mean SPI is <=0 (Figure 5). However, despite the limited number of episodes at each site, Figures 5 and 6 also show evidence of changes in the nature of episodes of groundwater drought associated with anthropogenic warming. The total number of episodes of groundwater drought at the sites is limited, and in large part reflects the natural variability of precipitation deficits. However, at both sites there is an increase in the frequency of episodes of groundwater drought between the first and last periods of analysis, i.e. from 12 to 19 and from 9 to 16 at CH and DH respectively for the periods 1891-1932 and 1974-2015 (Figure 6a and Tabls S3). Although note that at DH there were only 8 droughts in the middle analysis period 1933-1973, one less than in the first period. A year-long episode of groundwater drought started in December 1973 and ended in November 1974 at DH. This has been included in the statistics for the last analysis period. It illustrates the naturally 'noisy' nature of the relatively sparse data, reflects in part the temporal variability in the precipitation deficits that drive the occurrence of groundwater droughts at the site, and illustrates why we have chosen to analyse relatively coarse periods and use averages to characterise changes in drought characteristics across the record.

There is no consistent change in the mean duration of groundwater droughts at either CH or DH, with mean durations of about 11, 12 and 10 months across the three periods from 1891 to 2015 at CH, and mean durations at DH of 14, to 17, to 17 months for the same three periods. Although there is no clear tendency of change in mean $SGI_e$ duration, it appears that there may be a tendency for an increase both in the maximum drought duration and in the number of sub-annual episodes of groundwater drought, particularly at CH (Figure 6c). There is a tendency for the mean event magnitude and mean event intensity of groundwater droughts at both sites to increase with time, with mean event magnitude increasing more at DH that at CH, from about -12 to about -18, and mean event intensity to increasing more at CH than at DH, from -0.8 to -1.0 between the periods 1891-1932 and 1974-2015 (Figure 6b). The systematic increases in drought frequency, magnitude and intensity are associated with a large increase in mean STI across the three analysis periods, and are reflected in the change in relative position of the SPI-STI centroids of the data for $SGI_e$, shown as black stars, in the plots for each of the three periods in Figure 5.

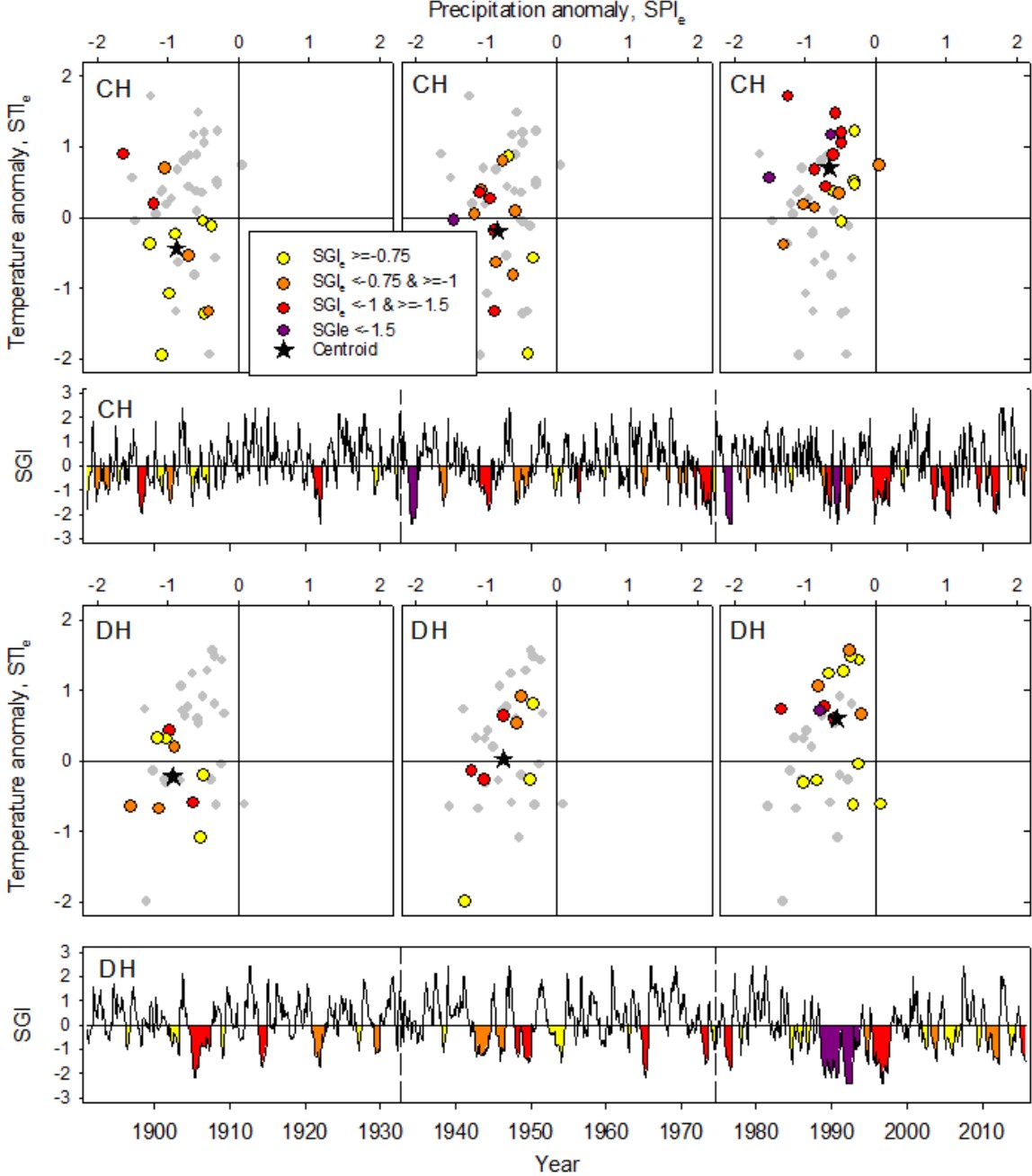

**Figure 5. Changes in the incidence and magnitude of episodes of groundwater drought since 1891 as a function of**
**temperature and precipitation. Mean groundwater drought event intensity (mean $SGI_e$) as a function of mean event**
**SPI and STI for the periods 1891-1932, 1933-1943, and 1944-2015. Grey symbols indicate all episodes of groundwater**
**drought at a site. Yellow through to purple symbols indicate increasing mean SGI for the episodes of groundwater**
**drought. The black star denotes the centroid of the episodes of SGI in each third of the observational record. The SGI**
**time series are shown below the cross plots for reference with SGI drought events of a given magnitude highlighted.**
**Data for CH is shown in the upper panels and DH in the lower panels.**


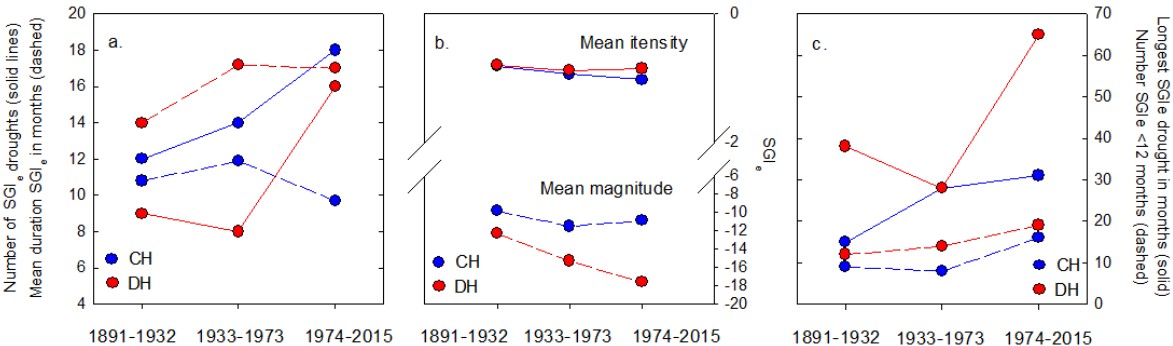


**Figure 6. a.** Change in total number of groundwater drought episodes and mean duration of SGI$_e$ at CH (blue) and
DH (red) for the periods 1891-1932, 1933-1973 and 1974-2015. **b.** Change in mean intensity and mean magnitude of
groundwater drought episodes over the three analysis periods at CH and DH. **c.** Change in maximum duration of
SGI$_e$ and number of episodes of groundwater drought less than 12 months in duration.

In addition to changes in mean STI associated with episodes of groundwater drought, there have been changes
in the coincidence of episodes of hot periods (STI$_e$) with episodes of groundwater drought (SGI$_e$) (Figure 7).
Episodes of groundwater drought generally coincide with precipitation droughts. Typically in a given period
precipitation droughts are co-incident with between 75% and 89% of groundwater droughts with no systematic
change in the frequency of co-incidence across the observational record at either site. This is consistent with the
observation that groundwater droughts at the sites are primarily driven by episodes of precipitation deficit.
However, there has been a large, systematic increase in the coincidence of groundwater droughts with hot
periods, from 17% to 29% to 79% at CH and from 33% to 50% to 63% at DH during the 1891-1932, 1933-1973
and 1974-2015 periods respectively (Figure 7).

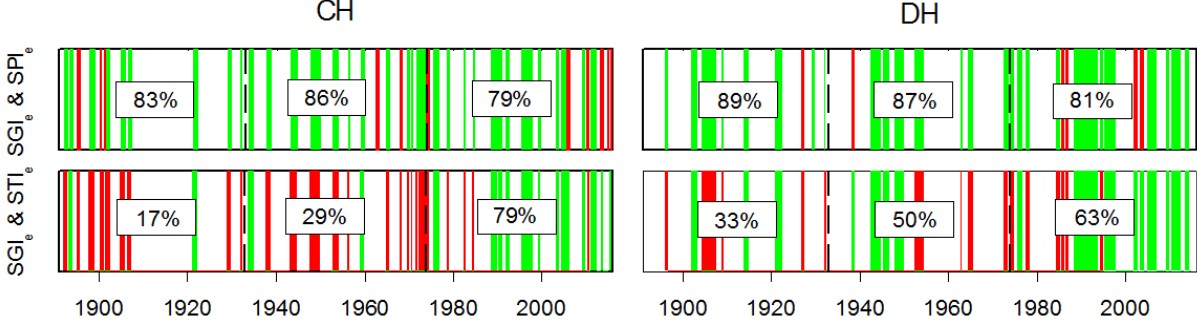


**Figure 7.** Graphical representation of coincidence of groundwater droughts with precipitation droughts (SGI-SPI)
and with hot periods (SGI-STI) for CH (left panels) and DH (right panels), where green denotes coincident and red
non-coincident episodes. Percentages indicate the fraction of coincident episodes for the first (1891-1932), middle
(1933-1943) and last (1944-2015) thirds of the record (where periods are separated by vertical dashed lines).
**5 Discussion and conclusions**
**5.1 Controls on changes in groundwater drought on the Chalk aquifer**
Precipitation deficit is the primary driver of groundwater drought (Tallaksen & Van Lanen, 2004; Van Loon,
2015). This is confirmed for CH and DH where dry months (SPI ≤ 0) are a broad prerequisite for groundwater
drought months throughout the record at both sites (Figure 4) and where almost all episodes of groundwater
drought (SGI ≤ -1) are associated with negative precipitation anomalies (SPI ≤ 0) (Figures 5 and 7). However,
we have also shown that there is no significant difference in the probability of precipitation drought between the
beginning and end of the records at CH and DH (section 4.1) and that increases in groundwater drought
frequency, magnitude and intensity are not associated with any long-term increases in precipitation deficits.
Given these observations, and in the absence of major long-term changes in land cover and the absence of
systematic effects of abstraction in the catchments (section 2.1), what are the controls on the observed changes
in groundwater drought since 1891 at CH and DH?
Marsh et al (2007) and Marsh (2007) have previously noted that major hydrological droughts in the UK may
persist for at least a year, and often substantially longer – a common feature of major groundwater droughts
globally (Tallaksen & Van Lanen, 2004; Van Loon, 2015). This is confirmed in the present study with the mean
duration of groundwater droughts at CH being about 11 months and at DH being about 16 months (Table S3).
Marsh et al (2007) and Folland et al (2015) also noted that major hydrological droughts in the UK are almost
always associated with more than one consecutive dry winter. However, there is no evidence for systematic
changes in the frequency of consecutive dry winters either at CH or at DH across the observational record. If a
dry winter is defined as below average mean monthly SPI for the winter half year (October to March), there no
consistent changes in consecutive dry winter years across the two sites. At CH there have been 16, 12 and 11
consecutive dry winters years over the periods 1891-1932, 1933-1973, and 1974-2015 respectively and at DH
14, 16 and 18 consecutive dry winter years over the same periods. In addition, there appears to be no systematic
drying trend associated with the consecutive dry winters when they do occur. For example, the mean SPI for
episodes of consecutive dry winters at CH is -0.72, -0.89, and -0.78 for the periods 1891-1932, 1933-1973, and
1974-2015, while the corresponding mean SPI for consecutive dry winters at DH it is -0.52, -0.70, and -0.62.
Since there is no clear driver for change in the nature of groundwater drought at CH and DH related to
precipitation deficits, and given that there is a significant increase in air temperature associated with
anthropogenic warming across the observational record, we postulate that increased evapotranspiration (ET)
associated with anthropogenic warming is a major contributing factor to the observed increasing occurrence of
individual months of groundwater drought as well changes in the frequency, magnitude and intensity of
episodes of groundwater drought.
In shallow, unconfined groundwater systems ET contributes to the formation and propagation of groundwater
droughts (Tallaksen & Van Lanen, 2004; Van Lanen et al., 2013; Van Loon, 2015) in a complex, non-linear
manner. As part of a study investigating the connections between groundwater flow and transpiration
partitioning based on modelling of data from shallow North American aquifers, Maxwell and Condon (2016)
have shown that ET is water limited below about 5m (Maxwell and Condon, 2016, Figure 3). If this relationship
holds for CH and DH, ET should be expected to have a limited effect on groundwater drought formation and
propagation at the two sites because the depth to groundwater at CH and DH associated with episodes of
groundwater drought is typically in the range 35 to 45 m and 10 to 15 m below ground level respectively. Unlike
many other aquifers, the Chalk has a thick capillary fringe and due to the micro-porous nature of the matrix
remains saturated to at least 30 m above the water table (Price et al., 1993). This potentially enables ET to
support the propagation of groundwater droughts even when the water table falls below 5m. Ireson et al. (2009)
have shown how groundwater flow through the unsaturated zone of the Chalk is highly sensitive to fracture
distributions and characteristics and so may be expected to vary significantly between sites on the Chalk as
fracture characteristics vary spatially (Bloomfield, 1996). However, even though there is no data on the
thickness of the capillary fringe at CH or DH, it can be estimated with some confidence due to the remarkable
uniformity of the matrix of the Chalk across the UK (Price et al., 1993, Figure 3.3a; Allen et al., 1997, Figure
4.1.5). Saturation of the matrix of the Chalk is controlled primarily by the pore-throat size distribution of the
matrix, which is characteristically less than 1 micron across the Chalk. Such pore throat sizes can support
capillary pressure heads of 30m or more, and consequently it has been proposed that this corresponds to the
typical depth of capillary fringe in the matrix of the Chalk aquifer (Price et al., 1993; Allen et al., 1997). From
the above, we infer that ET may be expected to contribute to the formation and propagation of groundwater
droughts at sites on the Chalk, such as at CH and DH, with water tables at least down to 30 m below ground
level. Consequently, on aquifers such as the Chalk, groundwater drought formation and development may be
particularly sensitive to the effects of changes in ET, and hence to anthropogenic warming.

**5.2 Implications for the changing susceptibility to global groundwater droughts**
Given that the sites analysed here are representative of groundwater systems in temperate hydrogeological
settings, we infer that anthropogenic warming may potentially be modifying characteristics of groundwater
drought such as the frequency, magnitude and intensity of groundwater droughts globally wherever shallow,
unconfined aquifers are present in temperate environments. If groundwater droughts are changing in their
character due to anthropogenic warming and that these changes are mediated by ET (Maxwell and Condon,
2016), how important might this phenomena be globally?

The partitioning of ET into plant transpiration, interception, soil and surface water evaporation at the continental
to global scale is challenging, however, Good et al. (2015) have estimated that the majority of ET, about 64%, is
due to plant transpiration. At the global scale, there is currently limited understanding of how plants use
groundwater for evapotranspiration. In the first such global analysis, Koirala et al., (2017) modelled the spatial
distribution of primary production and groundwater depth and found positive and negative correlations
dependent on both climate class and vegetation type. Positive correlations, i.e. higher plant productivity
associated with high (shallower) groundwater tables, were generally found under dry or temperate climate class
conditions, whereas negative correlations were associated with high plant productivity but with lower (deeper)
water tables predominately in humid environments. When just the temperate climate class was considered,
grass, crop, and shrub vegetation types (similar to those found at CH and DH) were all associated with positive
correlations between vegetative production and groundwater level, with only forests showing negative
correlations. Fan et al. (2013) produced the first high resolution global model of depth to groundwater level
depth, and, based on a conservative estimate of the maximum rooting depths of plants of 3 m below ground
level, estimated that up to 32% of the global ground surface area has a water table depth or capillary fringe
within rooting depth. Based on the observations above, it is clear that globally shallow groundwater systems are
common, that in temperate environments shallow groundwater contributes to ET mediated by plant
transpiration, and as such may be an important process effecting groundwater drought formation and
propagation, and hence may be susceptible to changes due to anthropogenic warming. If the effect of future
anthropogenic warming on groundwater droughts and more generally ET process in areas of shallow
groundwater and/or thick capillary finges are to be modelled with any fidelity, there is clearly a need for a focus
on improvements in modelling ET processes in shallow groundwater systems (Doble and Crosbie, 2017).

**5.3 Conclusions**

• In the fifth IPCC Assessment of Impacts, Adaptation, and Vulnerability it was noted that 'there is no
evidence that … groundwater drought frequency has changed over the last few decades' (Jiménez Cisneros
et al., 2014, p.232). Here we provide the first evidence for changes in groundwater drought frequency,
magnitude and intensity associated with anthropogenic warming. This has been possible due to the
unusually long and continuous nature of the groundwater level time series and supporting meteorological
data that is available for the CH and DH sites.

• The observed increase in groundwater drought frequency, magnitude and intensity at CH and DH
associated with anthropogenic warming is inferred to be due to enhanced evapotranspiration (ET). This is
facilitated by the thick capillary fringe in the Chalk aquifer which may enable ET to be supported by
groundwater through major episodes of groundwater drought.

• By extrapolation, as shallow groundwater systems are common and since in temperate environments
shallow groundwater contributes to ET mediated by plant transpiration this may be a globally important
process effecting groundwater drought formation and propagation. Wherever droughts in shallow
groundwater systems and/or aquifers with relatively thick capillary fringes are influenced by ET it is
inferred that they may be susceptible to changes due to anthropogenic warming.

**6. Author contributions**

J.P.B. conceived and coordinated the project. A.A.M. prepared the SGI, SPI and STI data for analysis. B.P.M.
performed statistical analyses. J.P.B. wrote the paper with input from A.A.M. and B.P.M.

**7. Competing interests**

The authors declare that they have no conflict of interest.

**8. Acknowledgements**

We would like to thank our colleague Mengyi Gong for the change point analysis. This work was undertaken in
part with support from the Natural Environment Research Council (NERC), UK, through the UK Drought and
Water Scarcity Programme project 'Analysis of historic drought and water scarcity in the UK: a systems-based
study of drivers, impacts and their interactions' (NERC grant NE/L010151/1) also known as the Historic
Droughts Project and by the Groundwater Drought Initiative (GDI) project (NERC grant NE/R004994/1). The
paper is published with the permission of the Executive Director of the British Geological Survey (NERC).

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
