# Peer review of "Changes in groundwater drought associated with"

_Hydrology and Earth System Sciences, 2018_

## Referee Comment (RC1) · Anonymous Referee #1 · 26 Jul 2018

The manuscript by Bloomfield et al. "Increased incidence, duration and intensity of groundwater drought associated with anthropogenic warming" submitted to Hydrolog. Earth Syst. Sci. Discuss presents and analyses two long-time series (1981-2015) dataset of piezometric head collected in a chalk aquifer in UK. The analysis takes advantage from the absence of major groundwater abstractions in the two study areas. This allows for assuming that the observed changes of the phreatic levels are due to climate variations. The manuscript is within the scope of the Journal and it is certainly of interest for the Readers of HESSD. It is well written, in both presenting the research framework and previous literature and showing and discussing results. However, I have some concerns that in my opinion should be addressed before a possible publication

on HESS. 1. I found figure 4 (the core business of the work) very interesting. As pointed out by the Authors, SPI<0 appears to be a broad prerequisite for groundwater drought. Moreover, I agree that increasing groundwater drought is associated with increasing temperature. However, there are some anomalies shown in figure 4, which in my opinion should be furtherly investigated. There is an interesting difference between the second and the third period for the CH dataset: the majority of the most intense groundwater episodes during the second period (SGI < -2) occur for negative or close to zero STI, whereas in the third period positive temperature anomalies seem to play a fundamental role, as almost all of the drought episodes (both for SGI < -2 and -2 < SGI < -1.5) are associated to STI > 0 and SPI < 0. This difference is strange and should be somehow explained: why the aquifer differently reacts to meteorological forcing? 2. Addressing the previous point, please consider also the following: the Authors "postulate that increased evapotranspiration associated with anthropogenic warming is a major contributing factor to the observed increasing occurrence of individual months of groundwater drought as well increasing the frequency, duration and intensity of episodes of groundwater drought" (page 15, line 404), despite the phreatic surface is approximately 40 m and 15 m below the topographic surface at CH and DH, respectively. According to the Authors, the fundamental role played by the transpiration is favoured by the significant thickness of the capillary fringe. This could be a possible explanation. However, at DH (where the water table is much higher than at CH, potentially making the aquifer more sensitive to temperature changes), the increase in temperature occurs over the entire period. I would have expected to find also in the second period an increase of the groundwater drought episodes with respect to the first period. In my opinion, an explanation to this anomaly should be given. 3. One more thing on the capillary fringe. Please, quantify its thickness for both sites. 4. It is not clear to me the reason for using standardized indexes in the analyses. As Authors know very well, standardized indexes are related to frequency (pdf) analysis. Therefore, doubling one index (i.e. from -1 to -2) does not mean doubling the intensity of the anomaly. Temperature and precipitation data come from gridded dataset and, considering the

limited surface of the study areas, only one pixel has been considered. Therefore, why not use directly observed data of precipitation and temperature? 5. Page 8, line 241. "The optimal averaging/accumulation period was found to be 6 and 7 months for CH and DH respectively". If I have understood correctly, Pearson correlation coefficients shown in figure S4 between SGI and SPI have been computed for a range of SPI accumulation periods considering the current month (i.e. SPIn of March, for n= 1,. . .12 is put in relation with SGI computed in March). This means that the percolation time from ground surface to saturated zone is neglected, not considering possible delay time of the impact of precipitation anomaly on groundwater level anomaly. Please, justify this choice. 6. Figure 6: I would have standardized the cumulative frequency distribution. In this analysis, I'm interested in a possible shift of the duration probability distribution. Probably they are very similar for the two analysed periods.

Minor remarks 7. Please, change the order of the subplots in figures 2 and 3 putting SPI first, then STI and finally SGI (groundwater drought is a consequence of climate anomalies) 8. Figure 5. As in this case SGI refers to the mean over a given drought episode and not to the monthly SGI, please use another notation.

---

## Author Comment (AC1) · 31 Jul 2018

We would like to thank Anonymous Referee #1 for their helpful and insightful observations and comments. Here we respond to the six main review comments and two minor comments, and where appropriate include brief notes on how we propose to revise the manuscript. We believe that the proposed revisions will improve the manuscript markedly.

Response to review comment 1.

Figure 4 is a plot of changes in the incidence and magnitude of groundwater drought

months since 1891 as a function of temperature and precipitation indices. The figure provides a picture of changes in the relationships between the three standardised indices, SPI, STI and SGI based on monthly data and illustrates where SGI is <-1, <-1.5 and <-2. We think that some of the confusion in the interpretation of this plot is due to i.) the assumption that the SGI <-2 highlighted data points represent multiple drought episodes, and ii.) that the groundwater drought analysis is based on months where SGI is <-2. For example the Reviewer refers to "the majority of the most intense groundwater episodes during the second period ...", and points to the apparently contrasting pattern in the distribution of SGI <-2 anomalies between the second and third periods at CH. Figure 4 does not show drought episodes, rather it shows monthly values of groundwater drought status. Groundwater drought episodes are shown in Figure 5. In fact, in Figure 4 eight of the nine months where SGI is <-2 in the second period at CH are associated with a single episode of groundwater drought (the 1933-34 drought), compared with five episodes of drought in the last period at CH. We note here that each major episode of groundwater drought at CH and DH has its own distinct characteristics associated with variations in antecedent conditions and the timing of onset and end of a drought episode, as seen in Figure 5. In this context, the occurrence of SGI <-2 months is of interest from the perspective of understanding the development of individual drought episodes, and we have included some illustrative additional background information (below) on the 1933-34 drought at CH. However, analysis of individual events is outside the scope of the current paper and we have avoided such analysis in the text. Instead, the focus of the analysis of the monthly data in Section 3.2 (as described in lines 266 to 274) is on comparative changes in monthly groundwater drought status across the observation record where groundwater drought is defined by analogy to the WMO definition, i.e. SGI <-1 (lines 246 to 256).

In summary, the differences in monthly values of SGI <-2 between the second and third periods at CH reflect both the response of the groundwater systems to noise in the driving drought climatology and the underlying change effect due to anthropogenic warming. The changes in drought incidence across the whole record are best described by changes in SGI <-1 months.

Note that in response to review comment 4 (see below), we propose to simplify Figures 4 and 5 so that all symbol sizes are the same in these figures. Hopefully this will give a more balanced view of the SGI monthly data populations across the three periods. In addition, we will also add some brief text to Section 3.4 reflecting the points outlined above.

Additional background information - evolution of the 1933-34 drought at Chilgrove House.

The major drought in the second period of the CH record associated with eight of the nine months where SGI is <-2 is the 1933-34 drought at Chilgrove House. This was the single most intense episode of groundwater drought at that site in the entire record, with a mean event SGI of -1.77 (see also Figure 5). We have included a short description here of the evolution of the 1933-34 drought at Chilgrove House as background information.

The 1933-34 groundwater drought at Chilgrove House started in September 1933, developed during the first half of winter 1933-34, and was maintained as a major groundwater drought through until a rapid end in November 1934 (see attached Figure 1 below). As illustrated below, the drought started following a period of above average temperature but was established due to a major deficit in precipitation with effectively no recharge during the normal winter and spring seasons. Temperature anomalies during this period would not be expected to make any significant contribution to the formation and propagation of the drought so that they were negative is not so relevant to this episode. The drought continued through the summer associated with a period of warming anomalies, and ended in November with a major episode of rainfall.

Response to review comment 2.

The Reviewer observes that they "would have expected to find also in the second

period [at DH] an increase of the groundwater drought episodes with respect to the first period". However, depending on how groundwater droughts are identified, there has been either a small decrease or a small increase in groundwater drought episodes between the first and second periods at DH. At DH there have been 33 groundwater droughts between 1891 and 2015, nine in the first period, eight in the second, and 16 events in the last period. However, if we consider droughts where average SGI is <-1 there have been two, three, and four events respectively across the three periods (Figure 5 and Supplementary Information, Table S3). We also note that if monthly SGI values are considered rather than drought episodes, then there was a small increase in the percentage of months where SGI is <-1 between the first and second third of the record at DH, from 15.3% to 19.1% of the months.

We know that the change signals that we are characterising are relatively subtle compared to the variability of the driving meteorology (see response to review comment 1 above), and we have already noted the typically low signal to noise ratio of hydrological systems (see lines 40-43). Consequently, throughout the paper we have chosen to focus on the differences in drought characteristics between the first and last third of the record, in large part because the last third of the record (see lines 202-203) coincides with the period of greatest documented warming over the study area (Karoly and Stott, 2006). We suggest that the fact that there were only eight episodes of groundwater drought identified during the middle period of the DH record compared with nine in the first third is simply a reflection of the noise in hydrological drought signals rather than an absence of a change phenomenon associated with warming.

The Reviewer also observes that "the phreatic surface is approximately 40 m and 15 m below the topographic surface at CH and DH, respectively ... at DH (where the water table is much higher than at CH, potentially making the aquifer more sensitive to temperature changes)". As long as there is a capillary connection between the land surface/plant rooting depth and the capillary fringe, the capillary fringe should provide a means for direct transpiration from groundwater to take place. Our inference is that,

for both CH and DH, the capillary fringe is thick enough to facilitate this exchange (see also response to Comment 3 below) and that there should be no relative 'sensitivity' related to the depth of the unsaturated zone as long as the capillary fringe exceeds the typical unsaturated zone thickness at both sites, which we postulate that it does.

We don't propose to make any changes to the text in response to this review comment.

Response to review comment 3.

Unfortunately, the specific thickness of the capillary fringe at both sites is unknown. In addition, no direct measurements of matrix water content, matric potential, pore-size or pore-throat size distributions in the matrix of the Chalk are available for either of these sites. Notwithstanding this, the size of pore throats in the matrix of the Chalk is remarkably uniform across the UK (see for example Price et al., 1993, Figure 3.3a and discussion in Allen et al., 1997, Figure 4.1.5), being characteristically less than 1 micron. Such pore throat sizes correspond to pressure heads of more than 30 m and as Price et al. (1993) notes "at equilibrium the pore spaces of the Chalk matrix can be expected to be saturated to a height of about 30 m above the water table" (Price et al., 1993, p. 40). We state on page 16, lines 419 to 421 that "The Chalk, ... is a dual porosity-dual permeability aquifer with a relatively thick capillary fringe. Due to the micro-porous nature of the matrix it remains saturated to at least 30 m above the water table (Price et al., 1993; Ireson, 2009)". For clarification, we propose to extend the text slightly as follows: "Although the specific thickness of the capillary fringe at Chilgrove House and Dalton Holme is unknown, the size of pore throats in the matrix of the Chalk is remarkably uniform across the UK (Price et al., 1993, Figure 3.3a; Allen et al., 1997, Figure 4.1.5), being characteristically less than 1 micron. Such pore throat sizes correspond to pressure heads of more than 30 m, and it is reasonable to infer that the matrix of the Chalk remains saturated to at least 30 m above the water table at the two study sites."

Response to review comment 4.

There are a couple of reasons why we have used standardised indices in the analysis. Firstly, standardised indices such as the Standardised Precipitation Index, SPI, (McKee et al., 1993) and the Standardised Groundwater level Index, SGI, (Bloomfield and Marchant, 2013) have been developed to enable direct comparisons to be made between meteorological or groundwater droughts across multiple sites. Although we are only comparing the response of groundwater levels at two sites, we still think that the use of standardised indices is helpful. For example, there is a small but systematic difference in the rainfall totals between the two sites. Figure S3 in the Supplementary Information shows the precipitation time series for the two sites. CH has a mean monthly precipitation of about 83 mm while DH has a mean monthly precipitation of 58 mm. We believe that removing such systematic differences in mean behaviour (in this case of precipitation) by standardising the two series makes any comparisons between the two sites more compelling. Secondly, as part of the analysis of the monthly data we wanted to formally test for significant differences in measures of precipitation, temperature and groundwater levels between the first and last third of the records (see lines 265 to line 272). By working with the standardised indices which are normally distributed we were able to assume that each of the standardised time series were a realisation of temporally auto-correlated Gaussian random function and hence test for any deviation in the observations from this assumption. We could not have performed this test on non-transformed, non-standardised data.

We thank the Reviewer for reminding us that the standardized indexes "are related to frequency (pdf) analysis . . . [and that] doubling of one index . . . does not mean doubling the intensity of the anomaly". Throughout the text we have been careful to characterise changes in the various indices by talking about percentage changes in numbers of months or numbers of drought events above or below a given threshold. In Section 3.3, where we discuss changes in the episodes of groundwater drought, we describe absolute changes in mean event intensity between the first and last third of the records. However, in our commentary we only note the direction of change: we don't analyse the absolute magnitude of those changes. Consequently, we don't believe that the
inherent non-linear scaling of the standardised indices is an issue for the text. We do, however, note that in Figures 4 and 5, the symbols indicating monthly SGI (Figure 4) and mean event SGI (Figure 5) thresholds are distinguished by both colour and symbol size. The latter is unnecessary and we will change all symbols on these two plots to the same size.

Response to review comment 5.

As noted in Section 3.2 (lines 236-244) groundwater levels typically display temporally lagged responses to precipitation. As a result, when relating SPI to SGI, SPI is usually accumulated over some period to establish the optimal relationship between the two standardised indices (McKee et al., 1993; Bloomfield and Marchant, 2013; Van Loon, 2015). In the Chalk aquifer of the UK optimal accumulation periods have previously been documented for a variety of sites in the range four to 28 months (Bloomfield & Marchant, 2013; Bloomfield et al., 2015). The relationship between SPI and SGI is purely phenomenological, enabling one index (the SPI) to be scaled so that it can be then correlated with the other (SGI) providing a process-independent description of the time varying relationship between rainfall and groundwater level response.

We don't propose to make any changes to the text in response to this review comment.

Response to review comment 6.

As suggested, we have revised Figure 6 (see below). In addition, we will include the standardised cumulative frequency plot for the middle period 1933-1973 for both sites, and will add additional histograms to show the change in absolute durations of ground-water drought events. We believe that this enables us to describe changes in the nature of the duration of events more effectively. Analysing the data in this manner shows that there is a small apparent increase in the probability of groundwater droughts up to durations of about 12 months at CH, but that overall the revised analysis is consistent with the assertion of the Reviewer that there is no clear shift of the overall duration probability distribution across the two sites. This is because, although the maximum duration

events occur in the last third of the records at each site and there is a tendency for there to be more long duration (greater than 12 months) events later in each record, there is also an increase in the number of shorter episodes of groundwater drought later in each record too.

Consequently, we propose to make the following changes to the paper. We will add the revised Figure 6 (as described above); we will revise the text in Section 3.3 to reflect the observation that there is no clear shift in the duration probability distribution at the two sites; and, we will revise other references in the text related to drought duration in line with the above.

Minor remarks – response to review comments 7 and 8. We'll change the order of the sub-plots as suggested. We'll change the SGI notation on the graph to read 'Mean event SGI'

References

Allen, D. J., Brewerton, L. J., Coleby, L. M., Gibbs, B. R., Lewis, M. A., MacDonald, A. M., Wagstaff, S. J., and Williams, A. T.: The physical properties of major aquifers in England and Wales, British Geological Survey Research Report WD/97/34, Keyworth, UK, 1997

Bloomfield, J. P., and Marchant, B. P.: Analysis of groundwater drought building on the standardised precipitation index approach. Hydrol. Earth Syst. Sci., 17, 4769–4787, 2013

Bloomfield, J. P., Marchant, B. P., Bricker, S. H., and Morgan, R. B.: Regional analysis of groundwater droughts using hydrograph classification. Hydrol. Earth Syst. Sci., 19, 4327–4344, 2015.Ireson, 2009

Karoly, D., and Stott, P.: Anthropogenic warming of Central England Temperature. Atmospheric Science Letters, 7, 81-85, 2006

McKee, T. B., Doesken, N. J., and Kleist, J.: The relationship of drought frequency and

duration to timescales. Proc. 8th Conf. App. Clim. 17–22 January 1993, Anaheim, California USA. 1993

Price, M, Downing, R. A, and Edmunds, W. M.: The Chalk as an aquifer. In: (Downing, R. A., Price, M., and Jones, G. P., eds.) The Hydrogeology of the Chalk of North-West Europe. Oxford Science Publications, Oxford, UK. p. 35-58. 1993

Van Loon, A. F.: Hydrological drought explained. WIREs Water, 2, 359–392, 2015
* * *
[Figure]

**Fig. 1.** Context to the 1933-34 drought

---

## Referee Comment (RC2) · Anonymous Referee #2 · 7 Oct 2018

**Overview**

First of all, I would like to thank the authors for providing such an interesting manuscript. I really enjoyed reading that.

This manuscript investigates changes in groundwater droughts associated with anthropogenic warming with no significant trend in precipitation. For this, the authors applied three standardized indices of SGI, SPI, and STI to two interestingly unique groundwater level time series in the UK covering 1891-2015. The major conclusions are: (1) the first evidence for an increase in the frequency, duration, and intensity of groundwater drought in response to anthropogenic warming; and (2) Such increases are mainly

inferred to increases in ET.

General comments

As this research analyses long-term (125 years) groundwater levels, precipitation, and temperature to explain changes in the groundwater drought under global warming, the topic of the paper is relevant for publishing in Hydrology and Earth System Sciences Discussions (HESSD). The introduction section sufficiently provides info related to groundwater drought, the approach used, and the study objectives. Description of the study area, data, and methodology applied in this manuscript are explained well. the "Results" Section is well structured, adequate for the purpose of the study, and the figures and tables are very informative (particularly Fig. 4 and 5). The "Discussion and Conclusions" Section gives an interesting explanation for results obtained. Hence, I recommend this manuscript for publication after some revisions.

Specific comment

To improve the manuscript, I suggest the authors include or respond to the followings:

1. In Line 58-60, the authors mentioned that such analyses requisite needs no systematic changes in precipitation. Did the authors analyze the trends in precipitation at the site studied for different time steps: full length (1891-2015), first third (1891-1932), second third (1933-1973), and last third (1974-2015)? I asked this question because the authors explained about changes in temperature at the sites (that follow the Central England Temperature, Lines 147-152) and provided Figure S2 in supplementary materials (to confirm it). However, there is no such explanation or figure for precipitation referring to the sites studied. The authors only mentioned that annual mean precipitation shows no trends since 1766 and also no attribution of changes in it to anthropogenic factors (Lines 152-156). Referring to the country-scale precipitation is not support that there are not any systematic trends in precipitation at two sites, even considering a 5km * 5km grid for each. I think the manuscript needs to clarify this issue to be more compelling.

2. In Lines 175-176, the authors mentioned they considered three periods for their analyses (1891-1932, 1933-1973, and 1974-2015) because each of these periods cover a considerable groundwater drought episode. At first, there is almost no text about the second third period (1933-1973), but it is included in all figures and tables. Then, in my opinion, the authors need to do the regime shift analysis for STI to identify the change point of the temperature time series as the authors are primarily looking for anthropogenic warming effects. Finally, based on these changing points, the groundwater droughts and SPI should be investigated.

Minor remarks

1. Line 279, "blue" should be revised to "red"

2. Figure 3, the numbers in the x-axis (1, 2, and 3) needs to be referred to the first (1891-1932), second (1933-1973), and last (1974-2015) third periods in the caption or legend.

3. Line 379, please remove of "anomalies"

4. Line 403, please remove "the" in "given the that"

5. Line 431, it should be "(Maxwell and Condon, 2016)"

6. Line 454, it should be "(Doble and Crosbie, 2017)"

7. Figure S4 in the supplementary materials, please provide the name of the site to the corresponding plot, in caption or legend.

---

## Referee Comment (RC3) · Anonymous Referee #3 · 11 Oct 2018

The study of Bloomfield et al., presents empirical evidence for growing influence of groundwater drought driven by evapotranspiration associated with anthropogenic warming. This is done by studying the relationship between drought patterns in groundwater observations (at two locales without presumable impact of abstractions or land use) and modelled temperature and precipitation data with different statistical methods. The interesting and well-written paper addresses relevant scientific questions, namely investigating changes in drivers of groundwater drought patterns anthropogenic change in a compartment of the water cycle. The work and line of thought is well documented and referenced. By additionally covering two out of three scopes of the journal, appropriate for publication in HESS.

[Figure]

Despite the novelty of the results and importance of the conclusions of the study, I have some concerns, which I think should be addressed. In my view there are some open questions mainly with regard to the application of the statistical methodology.

(1) Given that temperature rises throughout the three periods, this also means that high temperatures will coincide with groundwater drought more often. That means that over average STI values, e.g. STI > 1 will in this setup automatically be more common during the third period both for drought and flood conditions in groundwater, which can be seen in Fig 2. How much of the increase in temperature-related groundwater droughts does this account for? Groundwater droughts due to temperature could have been just as frequent in the earlier periods, but due to the trend in the STI values, just below the hard detection threshold of STI > 1.

(2) When finding the highest correlating SGI to SPI aggregation periods, you get correlation coefficients between .7 and .8 at 6 and 7 months respectively. Even though these values are considerably high, showing the SPI/SGI on a cross-plot would reveal a considerable number of events where SPI does not predict SGI well. I wonder therefore, whether there is a bias in the aggregation period. The SPI with selected aggregation period (e.g 6 months) over time could become a worse predictor, such that droughts associated with precipitation deficit become rarer (as seen in the third period)? A longer aggregation period would possibly show a smaller change in precipitation-related droughts. When looking at the study by two of the authors (Bloomfield and Marchant, 2013), the same locales were used among others, but DH had a longer aggregation period of 10 months, while using a shorter, more recent time period. Has a shift in the recharge regime occurred, which has been observed in other locations? If this is the case, surely the driver also is due to changes in the hydrological cycle.

(3) It did not become clear to me from the method section of the paper what was done with STI and why. As I understand, at different aggregation periods correlation coefficients between SGI and STI were calculated. These are generally weak 0 - -.2

(Supplement) and have a minimum (absolute maximum) at around 6 months, meaning that generally cold spells lead to more recharge and vice versa. There is quite some uncertainty involved though, at these low correlations, the relationship will be positive almost as often as it is negative. Despite this, I agree that this is the expected general tendency, I wonder though if this 6 month aggregation is still valid in the case of extreme events. My expectation would be that this behaviour changes and that for droughts only relatively short periods of relatively hot weather is needed for severe entailing groundwater droughts. If this is the case/could be shown, the findings would be even more interesting.

(4) 30 meters and more of thickness in capillary fringe seems unusually high. In Ireson et al. (2009) data was modelled for two locations different from CH and DH. Are these representative for CD and DH locations?

**Technical comments**

L234ff: Clarify that the indices are calculated over the entire period.
L240-241: Put maximum correlations into text
L269-272: Unclear what is meant by "probability of the difference", please specify what has been done here. Statistical significance?
Fig2: Very information-dense. The percentage values mean different things in the different panels, it should be possible to clarify within the figure.
Fig 3: Instead of using integers 1-3 for periods, use the interval of years on the y-axis.
Fig 4: Add location to the figure (CH, DH) so it becomes clear directly what the reader is looking at. Additionally it would be beneficial to see which of the non-drought months come from the specified period.
Fig 6: Why not include the second period? I get the impression from Fig 5 that drought durations are not dissimilar for the second and third period, especially for CH.

L412-416: Difficult sentence to digest, not clearly conveyed what the conclusions of the paper by Maxwell and Condon (2016) are.
Supplement, FigS4: Add locations CH/DH to the figure.

**References**

Bloomfield JP, Marchant BP. 2013. Analysis of groundwater drought building on the standardised precipitation index approach. Hydrology and Earth System Sciences, 17: 4769-4787. DOI: 10.5194/hess-17-4769-2013.

Ireson AM, Mathias SA, Wheater HS, Butler AP, Finch J. 2009. A model for flow in the chalk unsaturated zone incorporating progressive weathering. Journal of Hydrology, 365: 244-260. DOI: https://doi.org/10.1016/j.jhydrol.2008.11.043.

---

## Author Comment (AC2) · 7 Nov 2018

We would like to thank Referee #2 for their review comments on the paper and appreciate the feedback and insights that they have provided.

Response to the Specific comments

Specific comment 1. We entirely agree with Referee #2 that "referring to the country-scale precipitation is not support that there are not any systematic trends in precipitation at two sites". However, we do feel that we have addressed this issue in the paper, although perhaps not sufficiently explicitly.

[Figure]

As mentioned in the introduction (Lines 42-43), it is notoriously difficult to characterise trends in hydrometric variables, such as precipitation time series, that may contain a range of frequency structures and noise within them. This is one of the reasons (though not the only one) why we have not adopted a formal trend analysis approach to the study and have instead taken the approach of looking for systematic changes between periods in the long records. In this context, at Lines 265 to 272 we describe the results of a test of the probability of the difference in the number of dry months in the periods 1891-1932 and 1974-2015 and show that there is no statistical difference in precipitation between the start and the end of the records at both sites.

To make our approach more explicit, we propose to revise the first para of the Methods section at Line 198 as follows: "Given the aim of the study, one approach would be to undertake formal trend analyses of the standardised groundwater level, air temperature and precipitation data to see how each of these variables change over time at each site, to try and identify break points, and to see if and how any trends or break points are correlated. However, given the difficulties in identifying and quantifying local trends in long hydrological time series (Wilby 2006; Watts et al., 2015) and given that we are not interested in absolute trends or break points but rather in any changes in correlations between standardised variables consistent with our prior knowledge of the effects of climate change (Trenberth et al., 2015), we instead follow the approach of DIffenbaugh et al. (2015). They investigated changing frequency of drought, as measured by the Palmer Modified Drought Index (PMDI), with standardised annual average precipitation and temperature anomalies and looked at changes in those variables and their relationships. Diffebaugh et al., (2015) chose to analyse their 100-year-long records in two halves. However, in this study three periods have been used for the analysis, 1891-1932, 1933-1973 and 1974-2015. This means that the last period, 1974-2015, coincides with the period of greatest documented anthropogenic warming over the study area (Karoly and Stott, 2006). In addition, the use of three periods for analysis provides more granularity in the description of changes in the standardised indices with time. One benefit of this approach is that the assumptions

that there is i.) no change in precipitation between the first and the last periods at the two study sites, and ii.) a systematic, significant increase in temperature (previously attributed to climate change, Sexton et al., 2004; Karoly and Stott, 2006; Jenkins et al., 2008; King et al., 2015 ) across all three periods can be tested explicitly (see Results, section 3.1)"

Specific comment 2. Referee #2 suggests that, rather than use three equal periods for the analysis of the standardised time series, time series analysis techniques could be used to search for change points in the standardised temperature record and that these change points could then be used to sub-divide the time series prior to the STI-SGI-SPI analysis. There are a number of reasons why we did not follow this approach.

Reiterating our remarks related to Specific Comment 1 (above): formal trend analysis, and in particular identification of meaningful change points in hydrological or climatological time series, is particularly challenging and highly sensitive to noise in the signals and such formal time series methods are not consistent with our stated approach of simply describing changes in frequency and nature of groundwater droughts across a period of known warming. As importantly, there is no evidence of clear change points from previous analyses of anthropogenic warming in the CET. For example, the whole period of investigation is subject to continuous long-term warming with many local periods of superimposed variation in temperature, see for example Figure 1, p.2 of Kalroly & Stott (2006) and Figure 1.4, p10 of Jenkins et al (2009).

However, to address the specific comment, we propose to include in the modified introduction to the Methods section (see above) a note to the effect that we have not used change point analysis to sub-divide the analysis periods, rather that we have simply investigated and characterised changes in frequency and nature of groundwater droughts across a period of known long-term warming. Referee #2 also noted that "there is almost no text about the second third period (1933-1973), but it is included in all figures and tables". This was a conscious decision to reduce the description and discussion of the results in the text. We originally included more description of the observations and results for the middle third of the records but it made the paper more wordy without changing or adding to the substance of the findings, discussion or conclusions of the paper. In the light of the Referees comments we have reviewed the text again but propose not to add any extra text as we still think additional text would be superfluous. As the Referee notes, the results for the middle period are all available for the readers in the figures and tables.

Response to the Minor Remarks

1. Line 279, "blue" should be revised to "red". Agreed. In addition, the figure does not show a running mean, rather the line is the mean of the standardised index for each third of the record. Consequently, the text will be amended to read "The red bold line shows the mean standardised index for each third of each record".

2. Figure 3, the numbers in the x-axis (1, 2, and 3) needs to be referred to the first (1891-1932), second (1933-1973), and last (1974-2015) third periods in the caption or legend. Agreed. The caption will be revised to read as follows: "Percentage of monthly STI, SGI and SPI as a function of six ranges of standardised values from $\leq$ -2 to $\geq$ 2 for the first (1891-1932), middle (1933-1973) and last (1974-2015) thirds of the records from CH and DH, denoted by columns 1, 2 and 3".

3. Line 379, please remove of "anomalies". Agreed, the change will be made.

4. Line 403, please remove "the" in "given the that". Agreed, the change will be made.

5. Line 431, it should be "(Maxwell and Condon, 2016)". Agreed, the change will be made.

6. Line 454, it should be "(Doble and Crosbie, 2017)". Agreed, the change will be made.

7. Figure S4 in the supplementary materials, please provide the name of the site to the corresponding plot, in caption or legend. Agreed, the site names will be added to the plots.

References

Diffebaugh, N.S., Swain, D.L., and Touma, D.: Anthropogenic warming has increased drought risk in California. PNAS, 112, 3931-3936, 2015.

Doble, R.C., and Crosbie, R.S.: Review: Current and emerging methods for catchment-scale modelling of recharge and evapotranspiration from shallow groundwater. Hydrogeology Journal, 25, 3-23, 2017.

Jenkins, G.J., Perry, M.C., and Prior, M.J.: The climate of the United Kingdom and recent trends. Met Office, Hadley Centre, Exeter, UK. 2008.

Karoly, D., and Stott, P.: Anthropogenic warming of Central England Temperature. Atmospheric Science Letters, 7, 81-85, 2006.

King, A.D., van Oldenborgh, G.J., Karoly, D.J., Lewis, S.C., and Cullen, H.: Attribution of the record high Central England temperature of 2014 to anthropogenic influences. Environmental Research Letters, 10, 054002, 2015.

Maxwell, R.M., and Condon, E.: Connections between groundwater flow and transpiration partitioning. Science, 535 (6297), 377-380, 2016.

Sexton, D.M.H., Parker, D.E., and Folland, C.K.: Natural and human influences on Central England Temperature. Met Office Hadley Centre Technical Note HCTN 46, Met Office Hadley Centre, Exeter, UK. 2004.

Trenberth, K.E., Fasullo, J.T., and Shepard, T.G.: Attribution of climate extremes. Nature Climate Change, 5, 725-730, 2015.

Watts, G., Battarbee, R.W., Bloomfield, J.P., Crossman, J., Daccache, A., Durance, I., Elliott, J.A., Garner, G., Hannaford, J., Hannah, D.M., Hess, T., Jackson, C.R., Kay, A. L., Kernan, M., Knox, J., Mackay, J., Monteith, D.T., Ormerod, S.J., Rance, J., Stuart, M.A., Wade, A.J., Wade, S.D., Weatherhead, K., Whitehead, P.G., and Wilby, R.L.: Climate change and water in the UK: past changes and future prospects. Progress in

Physical Geography, 39, 6-28, 2015.

Wilby, R.L.: When and where might climate change be detectable in UK river flows? Geophys. Res. Letts., 33, L19407, 2006.

---

## Author Comment (AC3) · 7 Nov 2018

We would like to thank Referee #3 for their review comments. We enjoyed working though the challenges that the reviewer posed and think that a revised paper will be improved as a result of their contribution.

Response to the Specific comments

Comment 1. Referee #3 states that given the rise in temperature throughout the three periods, high temperatures will necessarily coincide with groundwater droughts more often in the latter period (see Fig 4), and that we have not investigated how many of

the observed groundwater droughts are attributable to increases in temperature over time. We agree that the former is the case, and emphasise that this is the main point of the paper, i.e. to characterise the change in groundwater drought in the context of prior knowledge of anthropogenic warming. However, with regard to the latter comment we would like to take the opportunity here to re-iterate the aim of the work as described in Lines 76-81, namely: "We have not attempted to formally attribute any groundwater droughts to climate change. Rather, we follow the approach of Trenberth et al. (2015) and investigate how climate change may modify a particular phenomenon of interest. In our case, given the known centennial-scale anthropogenic warming over the UK described in section 2.2 (Sexton et al., 2004; Karoly and Stott, 2006; Jenkins et al., 2008), using an empirical analysis we address the question: how has the occurrence and intensity of groundwater drought, as expressed by changes in SGI, changed over the same period?". In short, we are searching for evidence of changes in groundwater drought incidence, duration and intensity given known climate warming and the absence of other major change factors, and we are not currently concerned with the formal attribution of individual groundwater drought episodes. In this context, establishing changes in the number and nature of droughts associated with warming (in the absence of systematic changes in rainfall deficits) in the third period is the key result of the paper. We are not for example, making any further inferences from Fig 4. We note that this approach is explicitly similar to one used in the analysis of Diffebaugh et al (2015).

Although simple in approach, we believe that such an essentially descriptive, empirical analysis is important because of the current lack of observations relating anthropogenic warming to groundwater systems, and to groundwater droughts in particular. For example, the IPCC noted as part of the Fifth Assessment (WGII AR5) "that there is no evidence that surface water and groundwater drought frequency has changed over the last few decades, although impacts of drought have increased mostly due to increased water demand" (Jiménez Cisneros et al., 2014). We believe that our work here directly addresses the very limited evidence base and provides for the first time evidence

for the impact of climate change on groundwater droughts. We propose to make no changes to the paper related to this comment.

Comment 2. Referee #3 raises a number of questions related to the relationship between SGI and SPI. a. Referee #3 states that "When finding the highest correlating SGI to SPI aggregation periods, you get correlation coefficients between .7 and .8 at 6 and 7 months respectively. Even though these values are considerably high, showing the SPI/SGI on a cross-plot would reveal a considerable number of events where SPI does not predict SGI well". We feel that there is some confusion here regarding the purpose of aggregating the SPI to compare with SGI. SPI is known to be a poor predictor on it's own of groundwater drought events or SGI (e.g. Kumar et al., 2016; Van Loon et al., 2017), and this is not our aim in this study. Rather our purpose here is to identify an accumulation period that enables us to compare SPI and SGI in a consistent manner across the whole record. b. Referee #3 also states that "A longer aggregation period would possibly show a smaller change in precipitation-related droughts". Fig S4 shows that cross-correlation coefficients are relatively insensitive to accumulation periods for periods greater than about 5 to 7 months and for the purposes of our study it would not make sense to use longer accumulation periods for the analysis particularly where they have lower cross-correlations. c. Referee #3 states that "When looking at the study by two of the authors (Bloomfield and Marchant, 2013), the same locales were used among others, but DH had a longer aggregation period of 10 months, while using a shorter, more recent time period. Has a shift in the recharge regime occurred, which has been observed in other locations?". Fig 4 (and also Fig 7h in Bloomfield & Marchant, 2013) shows that differences in cross-correlation for accumulation periods between 6 and ~12 months is very small. Consequently, we don't interpret the differences in significant accumulation periods between the present study and that of Bloomfield & Marchant (2013) to be indicative of a "shift in the recharge regime", but merely a reflection of the relative insensitivity of the cross-correlation to SPI accumulation period beyond ~ 6 months. Our interpretation and the lack of evidence for a "shift in the recharge regime" is reinforced by the observation that the cross-correlations estimated for the first and last thirds of the records shown in Fig S4 are very similar.

Based on the above comments we propose to make a small clarification to the text at Lines 241-242, as follows: "The maximum cross-correlation between SGI and SPI was found for SPI accumulation periods of 6 and 7 months for CH (0.77) and DH (0.78) respectively. In addition, the maximum cross-correlation between SGI and STI was found to for an STI averaging period of between 5 to 6 months at CH (-0.15) and between 5 to 7 months at DH (-0.35). As would be expected, the cross-correlation between SGI and STI is weaker than that of SGI and SPI, but in all cases Figure S4 shows that there is limited sensitivity to the SGI-SPI and SGI-STI cross-correlations once a maximum correlation has been achieved after an accumulation or averaging period of about 6 to 7 months. Consequently, in order to treat the SPI and STI standardised data in a consistent manner across the whole record at each site, 6 and 7 month common accumulation and averaging periods have been estimated at CH and DH respectively. Although SPI6 and STI6 have been estimated for CH and SPI7 and STI7 have been estimated for DH, for simplicity throughout the following reporting of results and discussions all references to SPI and STI are for those accumulation and averaging periods for each site". In addition, we propose to make a minor adjustment to Fig 4, as follows: we will colour the cross-correlations estimated for the first and last thirds of the records to highlight their similarity to each other and to the estimates based on the whole record and hence emphasise the insensitivity of the correlations to the period of data on which they are based.

Comment 3. We agree that the cross-correlations between SGI and STI are relatively low compared with the cross-correlations between SGI and SPI. However, this does not have any bearing on our analysis. We are not using SGI-STI relationships to understand individual recharge events or for forecasting purposes, rather we are simply characterising the relationship between the two variables across the whole record. Because we are interested in the relationships between SGI, SPI and STI and how they change across the record it is important that they are estimated for common periods

in a consistent manner. Consequently, based on the correlations illustrated in Fig S4 we believe that we have adequately justified the common accumulation and averaging periods of 6 and 7 months for CH and DH respectively. We have amended the text at Lines 241-242 (see response to Comment 2 above) to emphasise these points.

Although it is outside the scope of the present study, we agree with Referee #3 that there would be merit in looking at relationships between individual groundwater droughts and characteristics of the antecedent air temperatures. However, we expect that any relationships would be complex and non-linear functions of a range of factors including antecedent groundwater levels and precipitation. Comment 4. There is a real paucity of observational data to constrain the height of the capillary fringe in the Chalk. The value of 30 meters for the thickness of the capillary fringe cited in the paper (Lines 420-421) is based on the theory of Price et al. (1993). This value is widely accepted in the absence of systematic observations. For example, while developing their Chalk unsaturated zone model, Ireson et al. (2009) also assumed that "the matrix [in the unsaturated zone of the Chalk] will generally remain saturated by capillary forces" and modelled changes in Chalk unsaturated zone pore pressure and water content as a function of variations in fracture incidence and aperture. Ireson et al. (2009) modelled field data from two sites from the Chalk of the Pang-Lambourn catchment in the Chilterns. Although the site was on the same aquifer formationas CH and DH, i.e. the Chalk, we agree that there is no reason to expect that results from those sites should necessarily be representative of CH and DH.

We propose to modify the text at Line 420 to clarify these points as follows: "The Chalk, however, is a dual porosity dual permeability aquifer with a thick capillary fringe. Due to the micro-porous nature of the matrix, the matrix theoretically remains saturated to at least 30 m above the water table (Price et al., 1993). Consequently, in the Chalk it is proposed that ET contributes to the formation and propagation of groundwater droughts at sites with water tables at least down to 30 m below ground level. If so groundwater drought formation and development may be particularly sensitive to the

effects of changes in ET, and hence to anthropogenic warming. We note that there have been no systematic observations of this phenomena across the Chalk aquifer to date and the only detailed observational study of variations in unsaturated zone flow, water content and matric potential in the Chalk is that of Ireson et al. (2009). They also assumed that matrix in the unsaturated zone remained saturated, and explained their observations in terms of the weathering profile of the Chalk and specifically variations the frequency and aperture of fractures. Clearly, if changes in ET mediated by anthropogenic warming are contributing to changes in groundwater drought in the Chalk and other shallow groundwater systems, there is a need to characterise this phenomenon using new co-located long-term soil moisture, water potential and groundwater level observations (Huntington, 2006). This is something that should be addressed with some urgency if we are to better constrain the effects of warming on groundwater resources and on groundwater droughts into the 21st Century."

Response to 'Technical comments'

L234: Referee #3 comment as follows: "Clarify that the indices are calculated over the entire period". Agreed. text to be modified to read as follows: "A Standardised Temperature Index (STI) and Standardised Precipitation Index (SPI) have been calculated by applying the SGI method to the average monthly temperature (STI) and a monthly accumulated rainfall (SPI) time series over the entire observation period."

L240-241: Referee #3 comment as follows: "Put maximum correlations into text". Agreed. See new text proposed in response to Comment 2 (above).

L269-272: Referee #3 comments that it is "unclear what is meant by "probability of the difference", please specify what has been done here. Statistical signiïficance?". The probability of difference can be thought of in the following way. If we define D as equal to the number of droughts in the last period minus the number of droughts in the first period (for example, but it could be any pair of periods) then the "probability of difference" is the probability, under the null model, that D is greater than the observed

value. To clarify the text we propose to revise it at Lines 266-269 as follows: "Given that the standardised indices are normally distributed, a null model can be estimated where each standardised index is assumed to be a realisation of temporally auto-correlated Gaussian random function (with auto-correlation function estimated from the observed data). A 'probability of difference' for a standardised index between periods can be estimated as follows: if we define D as equal to the number of droughts in the last period minus the number of droughts in the first period (for example) then the probability of difference is the probability, under the null model, that D is greater than the observed value. Estimated in this way, the probability of the difference in the number of hot months . . .".

Fig 2: Referee #3 comment as follows: "Very information-dense. The percentage values mean different things in the different panels, it should be possible to clarify within the figure". Agreed. The figure will be modified to be explicit regarding the %age exceedence in each figure.

Fig 3: Referee #3 comment as follows: "Instead of using integers 1-3 for periods, use the interval of years on the y-axis". We tried this in an earlier iteration of the figure however the text becomes too small to be legible at any sensible scale of reproduction. A similar comment was made by Referee #2. In response their comment and this comment we propose to modify the figure caption as follows: "Percentage of monthly STI, SGI and SPI as a function of six ranges of standardised values from $\leq$ -2 to $\geq$ 2 for the first (1891-1932), middle (1933-1973) and last (1974-2015) thirds of the records from CH and DH, denoted by columns 1, 2 and 3". In addition, we will label the x-axes as "Periods 1, 2 and 3".

Fig 4: Referee #3 comment as follows: "Add location to the figure (CH, DH) so it becomes clear directly what the reader is looking at." Agreed, the figure will be amended as suggested.

Fig 4: Referee #3 comment as follows: "Additionally, it would be beneficial to see which

of the non-drought months come from the specified period". Currently, the grey closed symbols denote all months data across all three periods. We will revise the figure to show this data as open symbols and use grey closed symbols to show non-drought months for the specific period.

Fig 6: Referee #3 comment as follows: "Why not include the second period? I get the impression from Fig 5 that drought durations are not dissimilar for the second and third period, especially for CH". Referee #1 raised a similar point. In response to their comment we have proposed to include the middle period in Fig 6 and make the following additional changes: "we will revise the text in Section 3.3 to reflect the observation that there is no clear shift in the duration probability distribution at the two sites; and, we will revise other references in the text related to drought duration in line with the above" (Bloomfield et al., 2018).

L412-416: Referee #3 comment as follows: "Difficult sentence to digest". We agree that the phrasing is clumsy and propose to re-draft as follows: "Based on analysis of data from shallow North American aquifers, Maxwell and Condon (2016) described a transition from a regime where T is groundwater dependent and E is water limited, to regime where both E and T are water limited. Under the latter regime groundwater is effectively disconnected from the land surface resulting in relatively low T and E that are limited by precipitation. They estimate that the transition between these regimes is of the order of 5 m below ground level".

Supplement, Fig S4: Referee #3 comment as follows: "Add locations CH/DH to the figure". Agreed, location details will be added to the figures.

References

Bloomfield, J. P., Marchant, B. P., Bricker, S. H., and Morgan, R. B.: Regional analysis of groundwater droughts using hydrograph classification. Hydrol. Earth Syst. Sci., 19, 4327–4344, 2015. Diffebaugh, N. S., Swain, D. L., and Touma, D.: Anthropogenic warming has increased drought risk in California. PNAS, 112, 3931-3936, 2015.

[Figure]

Huntington, T.G.: Evidence for intensification of the global water cycle: Review and synthesis. Journal of Hydrology, 319, 83-95, 2006

Ireson, A.M., Mathias, S.A., Wheater, H.S., Butler, A.P. and Finch, J.: A model for flow in the chalk unsaturated zone incorporating progressive weathering. Journal of Hydrology, 365, 244-260, 2009. Jenkins, G. J., Perry, M. C., and Prior, M. J.: The climate of the United Kingdom and recent trends. Met Office Hadley Centre, Exeter, UK. 2008.

Jiménez Cisneros, B. E., Oki, T., Arnell, N. W., Benito, G., Cogley, J. G., Döll, P., Jiang, T., and Mwakalila, S.S.: Freshwater resources. In: Climate Change 2014: Impacts, Adaptation, and Vulnerability. Part A: Global and Sectoral Aspects. Contribution of Working Group II to the Fifth Assessment Report of the Intergovernmental Panel on Climate Change. Cambridge University Press, Cambridge, United Kingdom and New York, NY, USA, pp. 229-269. 2014.

Karoly, D., and Stott, P.: Anthropogenic warming of Central England Temperature. Atmospheric Science Letters, 7, 81-85, 2006.

Kumar, R., Musuuza, J.L., Van Loon, A.F., Teuling, A.J., Barthel, R., Broek J.T., Mai, J., Samaniego, L., and Attinger, S.: Multiscale evaluation of the Standardized Precipitation Index as a groundwater drought indicator. Hydrol. Earth Syst. Sci., 20, 1117–1131, 2016

Maxwell, R. M., and Condon, E.: Connections between groundwater flow and transpiration partitioning. Science, 535 (6297), 377-380, 2016.

Price, M, Downing, R. A., and Edmunds, W. M.: The Chalk as an aquifer. In: (Downing, R. A., Price, M., and Jones, G. P., eds.) The Hydrogeology of the Chalk of North-Wwest Europe. Oxford Science Publications, Oxford, UK. p. 35-58. 1993.

Sexton, D. M. H., Parker, D. E., and Folland, C. K.: Natural and human influences on Central England Temperature. Met Office Hadley Centre Technical Note HCTN 46, Met

Office Hadley Centre, Exeter, UK. 2004.

Van Loon, A.F., Kumar, R., and Mishra. V.: Testing the use of standardised indices and GRACE satellite data to estimate the European 2015 groundwater drought in near-real time. Hydrol. Earth Syst. Sci., 21, 1947–1971, 2017
* * *

---

## Author Response (AR1)

**Response to reviewer's comments**

We would once again like to thank the Referees for their time and their thoughtful comments and insights. We believe that the review process has resulted in significant improvements to the paper.

**General observations**

The Editor has asked for the paper to be revised as follows:

> "The referee's comments were quite relevant, as it was also recognised by the authors in their replies. The answers to all the comments are interesting and deserve to be included somehow in the revised version of the manuscript. I expect that the authors can significantly improve the scientific quality of the paper by revising the text and the figures. In their replies, the authors propose to make no changes to the text in response to some of the referees' comments. I recommend the authors to carefully consider those comments and to evaluate whether rephrasing or reorganization of the text might avoid any possible misunderstanding by the readers".

Based on this feedback from the Editor, we have made modifications to the text and figures in response to **all** the comments from the Referees.

In addition to our responses to each of the referees specific comments (see below), we would like to highlight some changes to the data analysis in the revised version of the paper that have been stimulated by comments from the reviewers:

- Based on our response to Comment #5 from Referee #1 and Technical Comment #2 from referee #3, we present a more explicit description of the accumulation and averaging periods of the SPI and STI data, and in the revised paper we now use an accumulation period of six months, $SPI_6$, for Dalton Holme. This provides a more consistent treatment of the two sites while resulting in no changes to the main findings, inferences or conclusions of the study.
- In response to Comment #1 from Referee #2 and Comment #2 from Referee #3, we have provided more information on the statistical tests for change in the SGI, STI and SPI series, including notes on an additional change point analysis that we have undertaken. In so doing we found a minor error in our initial estimation of the level of significance of the change in STI between the first and last analysis periods (previously estimated to be <0.001 for both sites, and now corrected to read 0.03 for Chilgrove House and 0.005 for Dalton Holme). Again, these additions and changes, while adding more information, do not change the overall findings, inferences or conclusions of the study.
- In response to a number of comments, but particularly Comment #6 from Referee #1, we recognised that the old Figure 6 and the associated analysis in the text inadequately described changes $SGI_e$ characteristics between the analysis periods. Figure 6 in the original draft only described changes in drought duration, the revised new Figure 6 and associated revised text now describe both changes in the number of events, their duration, intensity and magnitude. As with the other changes to the manuscript, this adds more information but the overall findings and conclusions of the study remain unchanged.

Also, please note that we have taken the opportunity of the revision of the paper to simplify and amend text to improve legibility, for example we have simplified the title of the paper to "Changes in groundwater drought associated with anthropogenic warming". We have also corrected a number of typos, formatting errors (section number 3 was repeated in the original manuscript, so we have now re-numbered the sections accordingly), and some minor data transcription errors. Again, these changes do not affect the overall findings, inferences or conclusions of the study.

**Referee-specific responses and changes to the paper**

Note: All references to lines in the Referees Comments are to line numbers in the original paper unless otherwise stated. All references to lines in our Responses are to new line numbers in the revised paper unless otherwise stated.

**Referee #1**

Comment 1. "I found figure 4 (the core business of the work) very interesting. As pointed out by the Authors, SPI<0 appears to be a broad prerequisite for groundwater drought. Moreover, I agree that increasing groundwater drought is associated with increasing temperature. However, there are some anomalies shown in Figure 4, which in my opinion should be furtherly investigated. There is an interesting difference between the second and the third period for the CH dataset: the majority of the most intense groundwater episodes during the second period (SGI < -2) occur for negative or close to zero STI, whereas in the third period positive temperature anomalies seem to play a fundamental role, as almost all of the drought episodes (both for SGI < -2 and -2 < SGI < -1.5) are associated to STI > 0 and SPI < 0. This difference is strange and should be somehow explained: why the aquifer differently reacts to meteorological forcing?"

Response 1. Monthly values of SGI in the second period at CH are as much a response of the groundwater system to natural variability in precipitation deficits as they are to longer-term underlying changes associated with anthropogenic warming. The "most intense groundwater episodes during the second period (SGI < -2) [at CH]" noted by Referee#1 are in fact associated with a single major drought episode in 1933-1934. The effect of warming is better understood by considering the changing location of the centroid of the groundwater drought months (now added in the revised paper) in Figure 4.

To clarify these points we have:

- Added explanatory text at L402-411 to describe the effect of natural variability in precipitation deficits on data presented in Figure 4, including a note on the 1933-34 event.
- Added a new Figure S5 in the Supplementary Information illustrating how the intense groundwater drought months in the middle period at CH are associated with the evolution of the 1933-34 event.
- Added the centroid of the SGI data for each period to Figure 4 to explicitly show that the centroid of the SGI months is associated with an increasing mean STI with time and added new explanatory text at L425-430.

Comment 2. "Addressing the previous point, please consider also the following: the Authors "postulate that increased evapotranspiration associated with anthropogenic warming is a major contributing factor to the observed increasing occurrence of individual months of groundwater drought as well increasing the frequency, duration and intensity of episodes of groundwater drought" (page 15, line 404), despite the phreatic surface is approximately 40 m and 15 m below the topographic surface at CH and DH, respectively. According to the Authors, the fundamental role played by the transpiration is favoured by the significant thickness of the capillary fringe. This could be a possible explanation. However, at DH (where the water table is much higher than at CH, potentially making the aquifer more sensitive to temperature changes), the increase in temperature occurs over the entire period. I would have expected to find also in the second period an increase of the groundwater drought episodes with respect to the first period. In my opinion, an explanation to this anomaly should be given."

Response 2. As with our response to Comment 1 from Referee 1 (above), we feel that we have probably not emphasised sufficiently that groundwater droughts, as expressed in the monthly SGI (Figure 4) and by the episodes of groundwater drought SGI$_e$ (Figures 6 and 7), are primarily controlled by natural variation in precipitation deficits. The effects of anthropogenic warming are secondary and superimposed on the natural variability in precipitation droughts. At DH the second period was on average slightly wetter than the first period, while the last period is slightly drier than the first (mean SPI 0.03, 0.05, and -0.09 for periods 1891-1932, 1933-1973 and 1974-2015 respectively). Consequently, that the middle analysis period at DH has one less episode of groundwater drought than the first period is not surprising. In addition, we note that a year-long episode of groundwater drought started in December 1973 and ended in November 1974 at DH. This has been included in the statistics for the last analysis period. If it had been included in the middle period then the number of droughts at DH would have been 9, 9, and 15 for periods 1891-1932, 1933-1973 and 1974-2015 respectively. This discussion illustrates the naturally 'noisy' nature of the groundwater drought data driven by noisy precipitation data, the difficulty of identifying subtle long-term trends imposed by warming, and emphasises why we have chosen to analyse relatively coarse periods and use averages to characterise change across the record.

As we feel that we did not sufficiently emphasise the role of naturally varying precipitation deficits in controlling the overall pattern of drought episodes at the two sites in the initial draft we have added text to emphasise this point in the Abstract at L10-11, and in the text at L402-405, and L445-447 and at L507-510. In addition, we have added text at L445-458 to reflect Referee's comment and the specific discussion points above.

Comment 3. "One more thing on the capillary fringe. Please, quantify its thickness for both sites".

Response 3. The thickness of the capillary fringe at both sites is unknown and, in addition, there are no direct measurements of matrix water content, matric potential, pore-size or pore-throat size distributions in the matrix of the Chalk for either of the sites. However, it is known that the capillary fringe in the Chalk is primarily controlled by the microporus characteristics of the matrix, which are remarkably uniform across the Chalk of the UK.

We have added text at L552-555 to clarify these points.

Comment 4a. "It is not clear to me the reason for using standardized indexes in the analyses. As Authors know very well, standardized indexes are related to frequency (pdf) analysis. Therefore, doubling one index (i.e. from -1 to -2) does not mean doubling the intensity of the anomaly".

Response 4a. There are two broad approaches to identifying and characterising droughts including groundwater droughts. Namely, standardised indices and threshold level approaches (see Van Loon, 2015 for a recent overview). To clarify why we have used standardise indices in favour of the threshold approach to characterise droughts we have added a new discussion of the pros and cons of each approach and a justification of our choice at L262-279 at the start of the Methods section. In addition, we have added a note at L307-309 to explain the non-linear nature of standardised indices.

Comment 4b. "Temperature and precipitation data come from gridded dataset and, considering the limited surface of the study areas, only one pixel has been considered. Therefore, why not use directly observed data of precipitation and temperature?"

Response 4b. The extent of the groundwater catchments at the two study sites is unknown, though expected to be only a few square kilometres in extent (see Figure 1). There are no rain gauge or temperature records that continuously cover the study period at the sites. This study has been part of a larger study to reconstruct precipitation and hydrological droughts back to 1890 (the Historic Droughts Project, see https://historicdroughts.ceh.ac.uk/ ). As part of that study the Met Office in the UK has reconstructed gridded precipitation and temperature records on a 5km by 5 km grid across the UK back to 1890. We have chosen to use that gridded data product in this study since there is no other consistent dataset to use and we consider that it is at an appropriate resolution given the probably limited nature of the groundwater catchments at the two sites.

We have revised the text at L210-220 to clarify these points.

Comment 5. Page 8, line 241. "The optimal averaging/accumulation period was found to be 6 and 7 months for CH and DH respectively". If I have understood correctly, Pearson correlation coefficients shown in figure S4 between SGI and SPI have been computed for a range of SPI accumulation periods considering the current month (i.e. $SPI_n$ of March, for n = 1, … 12 is put in relation with SGI computed in March). This means that the percolation time from ground surface to saturated zone is neglected, not considering possible delay time of the impact of precipitation anomaly on groundwater level anomaly. Please, justify this choice".

Response 5. The optimal accumulation period is based on a phenomenological relationship between two time series, i.e. an accumulated precipitation and a groundwater level time series. It is not based on any consideration of recharge processes. Estimating correlations between a standardised hydrological drought index and varying SPI accumulation periods to characterise the relationship between precipitation deficits and droughts is a well-established approach, see for example a recent paper by Barker et al. (2016) in HESS on streamflow droughts, or various papers on groundwater droughts by the current authors (Bloomfield et al., 2013; 2015; Marchant et al., 2018). However, to address the specific query above, if the optimal accumulation period for precipitation is 6 months, i.e. $SPI_6$, then the March SGI is optimally correlated with all rainfall accumulated over the previous six month period, i.e. from (previous) October to March. From a process perspective, this would mean that March SGI would reflect any recharge that occurred during the six-month period prior to March. So although this is not a process-based relationship we assert that "percolation time from ground surface to saturated zone" is not neglected at all.

The text in the Methods section has been significantly extended at L336-346 to clarify these points. (Please note that in the revised paper we now use $SPI_6$ for both CH and DH. A full explanation is give in our response to Technical Comment #2 from Referee #3 below).

Comment 6. Figure 6: "I would have standardized the cumulative frequency distribution. In this analysis, I'm interested in a possible shift of the duration probability distribution. Probably they are very similar for the two analysed periods."

Response 6. When considering this comment and reading the interpretative text associated with Figure 6 in Section 3.3 we realised that we had provided only a limited description of changes in drought event characteristics. In addition, when considering the data for the limited number of drought episodes in each analysis period we have subsequently decided that analysis of the distribution of their characteristics may not be appropriate. Consequently, we have drafted a new Figure 6 to show changes in the frequency of episodes and mean drought episode characteristics across the analysis periods. We have also extensively revised the text discussing the new Figure 6 at

L445-471 to describe and discuss the new figure and data. We believe that this new plot and associated text now better describe changes in the characteristics of $SGI_e$ across the three periods.

Comment 7. (Minor remarks). "Please, change the order of the subplots in Figures 2 and 3 putting SPI first, then STI and finally SGI (groundwater drought is a consequence of climate anomalies)"

Response 7. Figures 2 and 3 have been revised as requested.

Comment 8. (Minor remarks). "Figure 5. As in this case SGI refers to the mean over a given drought episode and not to the monthly SGI, please use another notation."

Response 8. Revisions have been made to Figure 5 (and Figure 7) to include new notation to denote event mean SGI, as $SGI_e$. The Methods section has also been revised to make the distinction between SGI and $SGI_e$ clear at L352-355 and the Results and Discussion text modified appropriately too.

**Referee#2**

Comment 1. "In Line 58-60, the authors mentioned that such analyses requisite needs no systematic changes in precipitation. Did the authors analyze the trends in precipitation at the site studied for different time steps: full length (1891-2015), first third (1891-1932), second third (1933-1973), and last third (1974-2015)? I asked this question because the authors explained about changes in temperature at the sites (that follow the Central England Temperature, Lines 147-152) and provided Figure S2 in supplementary materials (to confirm it). However, there is no such explanation or figure for precipitation referring to the sites studied. The authors only mentioned that annual mean precipitation shows no trends since 1766 and also no attribution of changes in it to anthropogenic factors (Lines 152-156). Referring to the country-scale precipitation is not support that there are not any systematic trends in precipitation at two sites, even considering a 5km * 5km".

Response 1. We believe that the assertion that precipitation does not show long-term trends at either site is addressed in the paper already. For example, Figure 2 qualitatively shows no long-term variation in monthly SPI with time or in mean monthly SPI over the three analysis periods. Figure 3 also illustrates no long-term variation in SPI. However, more importantly we explicitly test the hypothesis that there is no statistically significant change in SPI across the record. At old L265-272 (new L371-384) we describe the results of a statistical test of the probability of the difference in the number of dry months in the periods 1891-1932 and 1974-2015 and show that there is no statistical difference in precipitation between the start and the end of the records at both sites.

Given the Referee's comment, we acknowledge that the above points were probably not articulated clearly enough in the original draft, consequently, we have added text justifying the use of the sites and their associated characteristics at the start of the site description section, section 2, by adding new text at L99-104.

In addition, a new change point analysis added in response to Comment 2 from Referee #2 (see below) indicates that there are no significant change points in the standardised precipitation time series, as described at L233-248.

Comment 2a. "In Lines 175-176, the authors mentioned they considered three periods for their analyses (1891-1932, 1933-1973, and 1974-2015) because each of these periods cover a considerable groundwater drought episode. At first, there is almost no text about the second third period (1933-1973), but it is included in all figures and tables."

Response 2a. We note the comments regarding the lack of discussion of the middle period in the text and recognise that it would be useful to include some discussion of the middle period where appropriate. However, we also want to make the Results and Discussion sections to be esily digestible and not overwhelm the reader with too much data. So we have where appropriate included additional information and comments regarding the middle period, for example see L428-429, L460-462, L495-497 and L526-531. Of course, data on all three periods is available in the Supplementary Information in Tables S1 to S3.

Comment 2b. "Then, in my opinion, the authors need to do the regime shift analysis for STI to identify the change point of the temperature time series as the authors are primarily looking for anthropogenic warming effects. Finally, based on these changing points, the groundwater droughts and SPI should be investigated."

Response 2b. We understand the suggestion from Referee 2 that two (or more) periods could be defined by using formal trend analysis methods to identify a change point in the temperature time series and that any differences in groundwater droughts could be analysed between those two periods. However, since we assert that groundwater droughts are primarily driven by precipitation deficits and modified by the effects of anthropogenic warming, and given that anthropogenic warming is not typically characterised by discrete change points, but rather has increasingly effected the climatology of the UK since the late 1800s, we don't think a change point approach based on the STI record is appropriate to the aim of the study. We are not interested in attribution of groundwater droughts. As we mention in the Introduction, Diffebaugh et al. (2015) have addressed a very similar problem to us and developed the simple but we think elegant solution of dividing the observational record in equal time periods and characterising and exploring change in hydrological drought in terms of concomitant change in SPI and STI between the periods. We have chosen to follow their approach.

However, based on the suggestion of Referee #2, we have investigated the SPI, STI and SGI series for change points to provide an additional insight into the data. We have added significant new text describing the new change point analysis, the results and a justification of why this approach has not been used to define the analysis periods at L233-260

Minor remark 1. "Line 279, 'blue' should be revised to 'red'"

Response Minor remark 1. Change made.

Minor remark 2. "Figure 3, the numbers in the x-axis (1, 2, and 3) needs to be referred to the first (1891-1932), second (1933-1973), and last (1974-2015) third periods in the caption or legend".

Response Minor remark 2. Change made to legend.

Minor remark 3. "Line 379, please remove of 'anomalies'"

Response Minor remark 3. Change made.

Minor remark 4. "Line 403, please remove 'the' in 'given the that'"

Response Minor remark 4. Change made.

Minor remark 5. "Line 431, it should be '(Maxwell and Condon, 2016)'"

Response Minor remark 5. Change made.

Minor remark 6. "Line 454, it should be '(Doble and Crosbie, 2017)'"

Response Minor remark 6. Change made.

Minor remark 7. "Figure S4 in the supplementary please provide the name of the site to the corresponding plot, in caption or legend".

Response Minor remark 7. Change made.

**Referee#3**

Comment 1. "Given that temperature rises throughout the three periods, this also means that high temperatures will coincide with groundwater drought more often. That means that over average STI values, e.g. STI > 1 will in this setup automatically be more common during the third period both for drought and flood conditions in groundwater, which can be seen in Fig 2. How much of the increase in temperature-related groundwater droughts does this account for? Groundwater droughts due to temperature could have been just as frequent in the earlier periods, but due to the trend in the STI values, just below the hard detection threshold of STI > 1".

Response 1. It is explicitly not our aim to attribute groundwater drought episodes to anthropogenic warming. Hence, we are not trying to answer the question: "How much of the increase in temperature-related groundwater droughts does this [anthropogenic warming] account for"? The aim of the paper is to undertake the first empirical analysis to characterise changes in groundwater drought incidence, duration and intensity given known climate warming and the absence of other major change factors (see original L76-84). We have revised the last paragraph of the Introduction at L82-96 to emphasise this point.

In addition, Referee #2 suggests that "Groundwater droughts due to temperature could have been just as frequent in the earlier periods, but due to the trend in the STI values, just below the hard detection threshold of STI > 1". For clarification, the frequency of occurrence of the observed groundwater droughts is fixed based on our definition of groundwater droughts where for either drought months or episodes of groundwater drought SGI <-1 or $SGI_e$<-1. We are not suggesting that any episodes of groundwater drought are "due to temperature" associated with any given STI threshold. Instead, we identify the groundwater droughts, and asses how their characteristics differ between the three analysis periods in the context of STI and SPI (where STI shows significant changes and SPI does not). We have made some minor revisions to the Introduction at L52-68 to emphasise this point.

Comment 2. "When finding the highest correlating SGI to SPI aggregation periods, you get correlation coefficients between .7 and .8 at 6 and 7 months respectively. Even though these values are considerably high, showing the SPI/SGI on a cross-plot would reveal a considerable number of events where SPI does not predict SGI well. I wonder therefore, whether there is a bias in the aggregation period. The SPI with selected aggregation period (e.g 6 months) over time could become a worse predictor, such that droughts associated with precipitation deficit become rarer (as seen in the third period)? A longer aggregation period would possibly show a smaller change in precipitation-related droughts. When looking at the study by two of the authors (Bloomfield and Marchant, 2013), the same locales were used among others, but DH had a longer aggregation period of 10 months, while using a shorter, more recent time period. Has a shift in the recharge regime occurred, which has been observed in other locations? If this is the case, surely the driver also is due to changes in the hydrological cycle".

Response 2. There is no evidence for changes in the maximum correlation between SPI and SGI or between STI and SGI across the whole observation period. The correlations for SPI-SGI over accumulation periods up to 12 months are similar between the first third and the last third of the observational record (Figure S4). They are also characterised by an insensitivity to accumulation periods once they reach a maximum at about 6 months, particularly at DH. This explains why for small differences in the observational record analysed maximum accumulation periods of 6 months (this study) and 10 months (Bloomfield and Marchant, 2013) have been observed. We revised the text at L323-334 to reflect these observations.

Comment 3. "It did not become clear to me from the method section of the paper what was done with STI and why. As I understand, at different aggregation periods correlation coefficients between SGI and STI were calculated. These are generally weak 0 - -.2 (Supplement) and have a minimum (absolute maximum) at around 6 months, meaning that generally cold spells lead to more recharge and vice versa. There is quite some uncertainty involved though, at these low correlations, the relationship will be positive almost as often as it is negative. Despite this, I agree that this is the expected general tendency, I wonder though if this 6 month aggregation is still valid in the case of extreme events. My expectation would be that this behaviour changes and that for droughts only relatively short periods of relatively hot weather is needed for severe entailing groundwater droughts. If this is the case/could be shown, the findings would be even more interesting".

Response 3. Monthly STI is estimated using the same method of standardisation as is used for SGI (and SPI) and STI standardisation (like SPI and SGI standardisation) has been performed across the full record (see L311-313).

In order to investigate relationships between the driving meteorology and hydrological responses, such as changes in monthly groundwater levels, it is common practice to search for a significant accumulation period for precipitation since hydrological response to meteorological anomalies is typically lagged. This approach was established by McKee et al (1993) when they introduced the SPI approach. We have estimated average STI over varying periods and correlated this with monthly SGI and then used this along with the estimates of correlations between SPI and SGI based on various SPI accumulations to find the accumulation / averaging period that gives the highest absolute summed cross-correlation for the entire observational series (see L323-334). We do this because we want to treat both the precipitation and air temperature aspects of the driving meteorology in a consistent manner.

We agree that the correlation between SGI and STI estimated in this manner is relatively week (this is now made explicit at L327-329), and that there is uncertainty in the relationship between SGI and $STI_6$. However, we want to re-emphasise that in the study we are exploring relationships between the standardised variables, we are not ascribing causative or predictive skill to the correlations illustrated in Figure S4. However, we note that the six month optimal accumulation/averaging period that is used provides a good characterisation of hydrological droughts. For example, practice in more process-based studies of the Chalk (e.g. Folland et al., 2015) it is common to consider the previous winter half year climatology as an indicator of subsequent groundwater drought status. New text has been added to L341-346 to reflect this observation.

Finally, we agree that the relationship between intense hot periods and groundwater droughts would be worthy of investigation. However, since the aim of the paper is to look at the relationship between centennial-scale warming and changing drought characteristics, investigation of the effect of extreme heatwave is out of scope of the current study. We have modified the text at L92-93 to this effect.

Comment 4. "30 meters and more of thickness in capillary fringe seems unusually high. In Ireson et al. (2009) data was modelled for two locations different from CH and DH. Are these representative for CD and DH locations?"

Response 4. There is a real paucity of observational data to constrain the height of the capillary fringe in the Chalk. The value of 30 meters for the thickness of the capillary fringe cited in the paper is based on the theory of Price et al. (1993). This value is widely accepted in the absence of systematic observations. For example, while developing their Chalk unsaturated zone model, Ireson et al. (2009) also assumed that "the matrix [in the unsaturated zone of the Chalk] will generally remain saturated by capillary forces" and modelled changes in Chalk unsaturated zone pore pressure and water content as a function of variations in fracture incidence and aperture. Ireson et al. (2009) modelled field data from two sites from the Chalk of the Pang-Lambourn catchment in the Chilterns. Although the site was on the same aquifer formation as CH and DH, i.e. the Chalk, we agree that there is no reason to expect that results from those sites should necessarily be representative of CH and DH since they were primarily considering the effects of fracturing on unsaturated zone drainage and this may vary widely between Chalk sites.

The text in the Discussion has been extensively revised at L539-562 to reflect the discussion above.

Technical comment 1. "L234ff: Clarify that the indices are calculated over the entire period".

Clarifications at L281-282 and at L311-313.

Technical comment 2. "L240-241: Put maximum correlations into text"

Correlations have now been detailed at L324 and 328. Please note, previously in the first draft of the paper an optimal accumulation and averaging period of 7 months for DH was based on the maximum absolute summed cross-correlation coefficient for SGI-SPI and SGI-STI estimated to greater than two significant figures. However, when estimating the co-efficient to two significant figures the coefficient is the same for 6 months and 7 months. Hence in the revised paper, 6 months has been used optimal accumulation and averaging period for DH and all data and plots have been revised appropriately.

Technical comment 3. "L269-272: Unclear what is meant by "probability of the difference", please specify what has been done here. Statistical significance"?

Additional explanation of the statistical analysis is now at L371-378.

Technical comment 4. "Fig2: Very information-dense. The percentage values mean different things in the different panels, it should be possible to clarify within the figure".

Figure 2 revised as requested.

Technical comment 5. "Fig 3: Instead of using integers 1-3 for periods, use the interval of years on the y-axis".

Figure 3 revised as requested.

Technical comment 6. "Fig 4: Add location to the figure (CH, DH) so it becomes clear directly what the reader is looking at. Additionally it would be beneficial to see which of the non-drought months come from the specified period".

Figure re-drafted as requested.

Technical comment 7. "Fig 6: Why not include the second period? I get the impression from Fig 5 that drought durations are not dissimilar for the second and third period, especially for CH".

Figure 6 has been re-drafted (see response to Referee#1, comment 6)

Technical comment 8. "L412-416: Difficult sentence to digest, not clearly conveyed what the conclusions of the paper by Maxwell and Condon (2016) are".

Text simplified significantly at L541-543 to clarify.

Technical comment 9. "Supplement, FigS4: Add locations CH/DH to the figure".

Locations added.

**References**

Lucy, J., Barker, L. J., Hannaford, J., Chiverton, A., and Svensson C.: From meteorological to hydrological drought using standardised indicators. Hydrol. Earth Syst. Sci., 20, 2483–2505, 2016

[revised manuscript text omitted]